# Dicer ablation in Kiss1 neurons impairs puberty and fertility preferentially in female mice

Juan Roa [1,2,3,4,8,9] ✉, Miguel Ruiz-Cruz [1,2,3,4,8], Francisco Ruiz-Pino[1,2,3,4], Rocio Onieva[1,2,3,4], Maria J. Vazquez[1,2,3,4], Maria J. Sanchez-Tapia[1,2,3,4], Jose M. Ruiz-Rodriguez [1,2,3,4], Veronica Sobrino[1,2,3,4], Alexia Barroso[1,2,3,4], Violeta Heras[1,2,3,4], Inmaculada Velasco[1,2,3,4], Cecilia Perdices-Lopez[1,2,3,4], Claes Ohlsson [5], Maria Soledad Avendaño[1,2,3,4], Vincent Prevot [6], Matti Poutanen[5,7], Leonor Pinilla[1,2,3,4], Francisco Gaytan[1,2,3,4] & Manuel Tena-Sempere [1,2,3,4,7,9] ✉

Kiss1 neurons, producing kisspeptins, are essential for puberty and fertility, but their molecular regulatory mechanisms remain unfolded. Here, we report that congenital ablation of the microRNA-synthesizing enzyme, Dicer, in Kiss1 cells, causes late-onset hypogonadotropic hypogonadism in both sexes, but is compatible with pubertal initiation and preserved Kiss1 neuronal populations at the infantile/juvenile period. Yet, failure to complete puberty and attain fertility is observed only in females. Kiss1-specific ablation of Dicer evokes disparate changes of Kiss1-cell numbers and Kiss1/kisspeptin expression between hypothalamic subpopulations during the pubertal-transition, with a predominant decline in arcuate-nucleus Kiss1 levels, linked to enhanced expression of its repressors, Mkrn3, Cbx7 and Eap1. Our data unveil that miRNA-biosynthesis in Kiss1 neurons is essential for pubertal completion and fertility, especially in females, but dispensable for initial reproductive maturation and neuronal survival in both sexes. Our results disclose a predominant miRNA-mediated inhibitory program of repressive signals that is key for precise regulation of Kiss1 expression and, thereby, reproductive function.

Fertility is governed by multiple regulatory factors that impinge on the hypothalamic-pituitary-gonadal (HPG) axis to modulate, in most cases, the activity of GnRH neurons as the final output pathway for the brain control of reproduction[1]. These neurons are scattered through the preoptic area (POA) of the hypothalamus, projecting their axons to the median eminence where gonadotropin-releasing hormone (GnRH) is secreted to the portal system to stimulate the secretion of gonadotropins, luteinizing hormone (LH), and follicle-stimulating hormone (FSH), by the pituitary[2]. Acquisition of reproductive capacity occurs during the pubertal period, a phenomenon driven by an increase in the

[1]Instituto Maimónides de Investigación Biomédica de Córdoba, 14004 Córdoba, Spain. [2]Department of Cell Biology, Physiology and Immunology, University of Córdoba, 14004 Córdoba, Spain. [3]Hospital Universitario Reina Sofia, 14004 Córdoba, Spain. [4]CIBER Fisiopatología de la Obesidad y Nutrición, Instituto de Salud Carlos III, 14004 Córdoba, Spain. [5]Department of Internal Medicine and Clinical Nutrition, Institute of Medicine, Sahlgrenska Academy, University of Gothenburg, 40530 Gothenburg, Sweden. [6]Univ. Lille, Inserm, CHU Lille, Laboratory of Development and Plasticity of the Neuroendocrine Brain, Lille Neuroscience & Cognition, UMR-S1172, 59000 Lille, France. [7]Institute of Biomedicine, Research Centre for Integrative Physiology and Pharmacology, and Turku Center for Disease Modeling, University of Turku, 20520 Turku, Finland. [8]These authors contributed equally: Juan Roa, Miguel Ruiz-Cruz. [9]These authors jointly supervised this work: Juan Roa, Manuel Tena-Sempere. ✉e-mail: roarivas@gmail.com; fi1tesem@uco.es

activity of the GnRH pulse generator that dictates the pulsatile secretion of gonadotropins to stimulate gonadal maturation. This pattern of pulsatile secretion of GnRH and gonadotropins is also mandatory for the proper functioning of the reproductive axis thereafter. While numerous factors, including intrinsic properties of GnRH neurons[3], are involved in the maintenance of GnRH pulsatility, Kiss1 neurons have emerged in the last years as a major component of the so-called GnRH pulse generator[4].

Kiss1 neurons produce kisspeptins, a family of neuropeptides encoded by *Kiss1*, that stimulate GnRH secretion acting via the G-protein coupled receptor, Gpr54[1]. A wealth of data have conclusively documented that *Kiss1* and *Grp54* are essential for the acquisition and maintenance of reproductive capacity in mammals, including humans[1,5]; mice and humans harboring inactivating mutations of these genes suffer impuberism and hypogonadotropic hypogonadism. This essential role in the control of reproductive axis is conducted by discrete populations of Kiss1 neurons, which in rodents are mainly located in two different hypothalamic areas, namely the arcuate-nucleus (ARC) and the anteroventral periventricular nucleus (AVPV)[6]. While both populations produce kisspeptins, they display interesting neuroanatomical and functional divergences. Thus, the AVPV Kiss1 neuronal population is far more prominent in females and is activated by estrogens to mediate their positive feedback for the induction of the preovulatory LH surge. In contrast, the ARC Kiss1 neuronal population is present in both sexes and is repressed by gonadal steroids to convey their negative feedback actions essential for the tonic release of GnRH[7]. The molecular underpinnings for these differential features remain largely unknown.

ARC Kiss1 neurons have been shown to produce also other neuropeptides with essential roles in reproductive control, namely, neurokinin B (NKB) and Dynorphin (Dyn); hence, the term KNDy (for Kisspeptin, Neurokinin B, and Dynorphin) has been coined to name this neuronal population. In fact, KNDy neurons have been pointed to as the cornerstone of the GnRH pulse generator system[4], by virtue of the presence of para/autocrine regulatory mechanisms involving NKB, Dyn, and their receptors, to drive the activity of Kiss1 neurons in a pulsatile manner[8]. This regulatory network, which was initially formulated on the basis of expression analyses and electrophysiological data[8,9], has been recently corroborated by optogenetic photo-stimulation of KNDy neurons[10]. Likewise, GCaMP fiber photometry and optogenetics/pharmacogenetics in vivo have documented that KNDy neuron activation produces a self-priming increase of basal and episodic calcium events, correlating with pulsatile LH secretion[4], as a surrogate marker of GnRH bursts. These data conclusively document the key role of KNDy neurons within the GnRH pulse generator.

While our understanding of the physiology and regulation of Kiss1 neurons has expanded substantially[11–13], critical aspects of the molecular mechanisms whereby the development and function of Kiss1 neuronal populations are precisely controlled remain poorly characterized. Epigenetic regulatory mechanisms have recently emerged as putative modulators of puberty, through regulation of *Kiss1* expression via SIRT1-mediated modifications of the chromatin landscape[14]. In addition, changes in histone acetylation in the *Kiss1* promoter in AVPV vs. ARC neurons seemingly contribute to the differential feedback actions of estrogen at these sites[15]. However, the putative role of other non-transcriptional regulatory mechanisms in the precise control of Kiss1 neurons remains largely unexplored. MicroRNAs (miRNAs) are short non-coding RNA sequences that regulate a large proportion of human protein-coding genes, mainly by post-transcriptional repression[16,17]. Canonical biogenesis of miRNAs involves a concatenated series of nuclear and cytoplasmatic events, in which the RNAse III enzyme, Dicer, is mandatory for final generation of mature miRNAs, and initiates also the formation of the RNA-induced silencing complex (RISC), responsible for gene repression[18]. Over the last decades, miRNAs have been shown to participate in the regulation of a large number of physiological and pathological processes, ranging from somatic development to cancer progression[19,20]. Interestingly, several studies have suggested a major role of miRNA-regulatory pathways at different levels of the HPG axis, with distinct roles in the control of gonadal and gonadotrope maturation and function[21–24]. More recently, specific miRNAs have been involved in the control of key hypothalamic factors for the regulation of puberty, such as GnRH and Makorin3[25,26]. However, the putative physiological roles of miRNA-regulatory pathways in the control of the maturation and function of Kiss1 neurons in vivo have not been explored to date. By generation of a mouse line with conditional ablation of Dicer in Kiss1-expressing cells, named KiDKO, we provide herein conclusive evidence on the need for proper miRNA biosynthetic machinery in Kiss1 neurons for adult reproductive function, and surface also interesting differences, between sexes and Kiss1 neuronal subpopulations, in the roles of miRNA-regulatory pathways in Kiss1 neurons for the precise regulation of the brain networks governing reproduction.

## Results

### Selective ablation of Dicer in Kiss1 neurons: the KiDKO mouse

Mice with selective elimination of the miRNA maturing enzyme, Dicer, in Kiss1 cells, named KiDKO for Kiss1-specific Dicer KO, were generated using the Cre-LoxP strategy. KiDKO mice were born at a Mendelian ratio. Specific ablation of Dicer in Kiss1 neurons was assessed by PCR of the recombinant *Dicer* allele, in situ hybridization, and qPCR on FACS-isolated Kiss1 neurons. A recombinant *Dicer* allele was detected by PCR in peripubertal (PND28) KiDKO mice, on genomic DNA isolated from the hypothalamic regions containing Kiss1 neurons (namely, ARC and POA, the later including AVPV), while it was not detectable in the cortex (area without reported Kiss1 expression) of KiDKO mice, or in hypothalamic samples from control animals, used as negative control. This recombination event was detected as early as on PND1 in the hypothalamus of KiDKO but not of control mice (Fig. S1a). In addition, dual in situ hybridization, using BaseScope™ probes for detection of mRNAs corresponding to *Kiss1* and *Dicer* Exon 23 (i.e., the loxP-flanked region subjected to recombination), documented co-localization in controls but not in KiDKO mice, therefore confirming that Kiss1 neurons normally express Dicer, but this is ablated in KiDKO animals (Fig. S1b). Note that this proof-of-principle analysis was conducted in young-adult, ovariectomized KiDKO mice, as a means to enhance detection sensitivity. As an additional control, TaqMan qPCR analyses in ARC Kiss1 neurons isolated at puberty, using FACS in KiDKO mice expressing YFP in Kiss1 cells, demonstrated that the levels of an abundant miRNA, namely let-7b, were significantly blunted in KiDKO mice (Fig. S1c); effective isolation of Kiss1 neurons by our FACS protocol was confirmed by detectable expression of *Kiss1* and *Tac2*, but not of *Npy*, in Kiss1 cells obtained from the ARC of KiDKO and control mice (Fig. 1c). In the same vein, expression analyses in FACS cells demonstrated a dramatic drop in the percentage of Kiss1 neurons expressing *Dicer* in peripubertal KiDKO mice (Fig. S1d). Accordingly, *Dicer* expression was significantly diminished in the ARC of KiDKO males and females at PND14 and PND28, and in the AVPV of peripubertal KiDKO females (Fig. S1e). Of note, mice engineered to lack Dicer in GnRH neurons, named GoDKO (for Gonadotropin-releasing hormone-specific Dicer KO), were generated also as described in detail elsewhere[25], and were analyzed in parallel for comparative purposes in selected phenotypic studies.

### Hypogonadotropic hypogonadism and infertility in mature adult male and female KiDKO mice

Congenital ablation of Dicer in Kiss1-expressing cells caused a profound state of hypogonadotropic hypogonadism (HH) in male and female KiDKO mice of 4–6 months of age, i.e., mature adult period, following the standard nomenclature of the Jackson

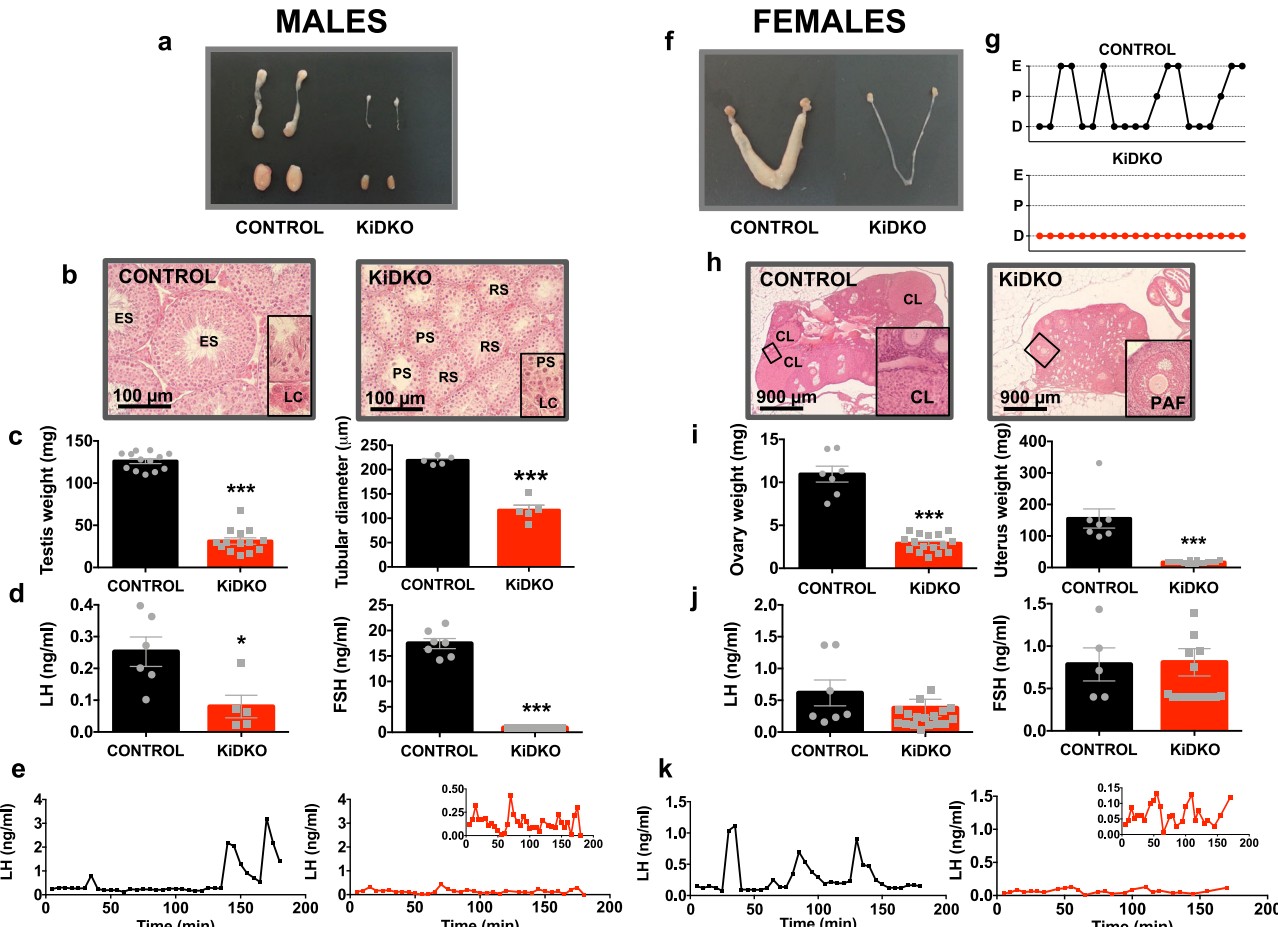

**Fig. 1 | Profound hypogonadotropic hypogonadism in adult KiDKO mice.**
Representative images of the gonads (testes and ovaries) and sex organs (epididymis and uterus) for adult control and KiDKO mice of both sexes are presented in (**a**; males) and (**f**; females). For the latter, individual profiles of ovarian cyclicity, monitored by vaginal cytology, are also shown in (**g**). In addition, representative histological sections of the testis (**b**) and ovary (**h**) are presented for control and KiDKO mice. Control mice showed complete spermatogenesis in the lumen of seminiferous tubules, denoted by the presence abundant elongated spermatids (ES) and prominent Leydig cells (LC; inset) in the interstitium. In contrast, KO mice showed atrophic seminiferous tubules with spermatogenic arrest at the primary spermatocyte (PS) or round spermatid (RS) level, as well as poorly differentiated Leydig cells (LC; inset). In the case of the ovary, while control females showed cycling ovaries with abundant growing follicles and corpora lutea (CL), in KiDKO females, large antral follicles and corpora lutea

were absent in KO mice, while preantral (PAF; see inset) or early antral follicles were the most advanced healthy growing follicles. In addition, testicular weight and tubular diameter in males (**c**; control $n = 13$, KiDKO $n = 14$ for testis weigh and control $n = 5$, KiDKO $n = 5$ for tubular diameter), and ovarian and uterus weight in females (**i**; control $n = 7$, KiDKO $n = 16$) are presented for control and KiDKO mice, for which serum LH and FSH levels are also shown (**d**; LH: control $n = 6$, KiDKO $\underline{n} = 5$; and FSH: control $n = 7$, KiDKO $n = 8$; **j**; control $n = 7$, KiDKO $n = 17$ for LH levels and control $n = 5$, KiDKO $= 18$ for FSH levels). Finally, pulsatile secretory profiles of LH over a 180-min period were assessed in control and KiDKO animals (**e**; males: control $n = 14$, KiDKO $n = 12$; **k**; females: control $n = 8$, KiDKO $n = 7$); only representative profiles of male and female mice from both genotypes are shown. Note that results for LH pulsatility in KiDKO mice are also presented at a magnified scale in the *insets*. The values are represented as the mean ± SEM. *$P < 0.05$; ***$P < 0.001$ vs. corresponding control groups.

Laboratory. This HH phenotype was denoted by a significant atrophy of the reproductive organs (Fig. 1a, f), with testis, uterus, and ovarian weights being clearly diminished in conditional null animals (Fig. 1c, i). Histological analysis of the gonads showed a decrease in the tubular diameter and absence of mature spermatids in the testes of adult KiDKO mice, while in conditional null females, the ovaries showed an absence of corpora lutea, denoting lack of ovulation (Fig. 1b, c, h). The latter was also confirmed by the absence of estrous cyclicity, with persistent diestrus in vaginal smears, and the presence of preantral follicles as the most advanced stage of healthy growing follicles (Fig. 1g, h). Hormonal analyses documented disturbed gonadotropin secretion and suppressed circulating sex steroid levels as proof of central hypogonadism in KiDKO mice. Thus, adult KiDKO male mice displayed diminished serum levels of LH and FSH in single time-point measurements (Fig. 1d), as well as significantly reduced serum testosterone concentrations (Table 1). Likewise, time-course analyses of LH pulsatility, assayed by ultrasensitive ELISA on serial blood samples,

documented suppressed basal LH levels in KiDKO male mice (Control: $0.14 \pm 0.02$ ng/mL vs. KiDKO: $0.07 \pm 0.01$; $p = 0.01$) and absence of normal LH peaks (Fig. 1e). Interestingly, although adult KiDKO female mice did not show significantly suppressed gonadotropin levels in single time-point measurements (Fig. 1j), or basal LH levels in pulsatility analyses (Control: $0.09 \pm 0.02$ ng/mL vs KiDKO: $0.07 \pm 0.01$; $p = 0.44$), LH peaks were also significantly blunted in female KiDKO mice (Fig. 1k). Yet, although the amplitude of LH peaks in KiDKO animals of both sexes was significantly reduced, peak frequency (#LH peaks/h) was not statistically different from controls (Males: Control: $0.85 \pm 0.31$ LH peaks/h vs KiDKO: $1.33 \pm 0.46$; $p = 0.39$; Females: Control: $2.12 \pm 0.58$ vs KiDKO: $0.85 \pm 0.45$; $p = 0.11$). In addition, adult KiDKO females displayed significantly reduced serum estradiol and progesterone levels, further attesting their hypogonadal state (Table 1).

The KiDKO phenotype largely recapitulated the features found in adult mice with congenital ablation of Dicer in GnRH neurons (Fig. S2).

**Table 1 | Circulating sex steroid levels in KiDKO and GoDKO male and female mice at two age windows: peripubertal (4-week-old) and adult (>2-month-old)**

| MALES | Peripubertal (4-wk-old) | | | Adult (>2 month-old) | | |
|---|---|---|---|---|---|---|
| | Control n = 17 | KIDKO n = 5 | GODKO n = 4 | Control n = 43 | KIDKO n = 17 | GODKO n = 5 |
| Testosterone (pg/mL) | 84.96 (25.26, 338.52) | 177.48 (60.70, 225.98) | 107.32 (63.68, 455.13) | 440.93[#] (106.97, 17379.55) | 82.17[a] (5.80, 1635.95) | 5.00[a,#,1] (5.00, 5.95) |
| Progesterone (pg/mL) | 273.60 (119.90, 584.66) | 246.43 (192.80, 302.93) | 141.82[a] (119.10, 185.60) | 122.00[#] (43.70, 401,43) | 114.23 (43.19, 525.14) | 41.98[a,b,#] (37.27, 93.32) |
| FEMALES | Peripubertal (4-wk-old) | | | Adult (>2 month-old) | | |
| | Control n = 28 | KIDKO n = 11 | GODKO n = 3 | Control n = 36 | KIDKO n = 13 | GODKO n = 4 |
| Estradiol (pg/mL) | 0.51[2] (0.50, 17.70) | 0.50[3] (0.50, 2.77) | 0.50[4] (0.50, 0.59) | 1.76[#] (0.5, 14.26) | 0.50[a] (0.50, 0.71) | <0.50[a,5] |
| Progesterone (pg/mL) | 169.98 (93.64, 343.19) | 174.07 (110.77, 262.62) | 157.84 (131.37, 198.99) | 492.38[#] (70.97, 8262.06) | 44.11[a,#] (30.42, 120.46) | 104.54 (74.34, 252.68) |

For males, testosterone and progesterone levels are presented; for females, estradiol and progesterone concentrations are shown.
Data were expressed as Median (min, max). For data analysis, samples below the limit of quantitation (LOQ) of each analyte were assigned the LOQ value (Testosterone: 5 pg/mL; Estradiol: 0.5 pg/mL).
[#]Statistical differences between adults and peripubertal animals of the same genotype ($p < 0.05$).
[a]Statistical differences vs age-matched controls ($p < 0.05$).
[b]Statistical differences between GoDKO and age-matched KiDKO mice ($p < 0.05$).
[1]Three out of five samples <LOQ.
[2]Thirteen out of 17 samples <LOQ.
[3]Nine out of 11 samples <LOQ.
[4]Two out of three samples <LOQ.
[5]All (four out of four) samples <LOQ.

GoDKO mice displayed a state of HH, denoted by atrophy of gonadal structures in both sexes, and accompanied by the absence of any sign of ovulation in females and lack of mature spermatids in males (Fig. S2a–c, e–h). Basal LH and FSH levels, as well as circulating sex steroids, were significantly reduced in both males and females at adulthood (Fig. S2d, i, and Table 1).

Comparative analyses were conducted in KiDKO and GoDKO mice to assess the differential impact of Dicer ablation in these two key neuronal populations in terms of neuropeptide gene expression and neuroendocrine responses to major stimuli. In line with a discrete pattern of Dicer elimination in both lines, adult KiDKO mice displayed a preserved number of *GnRH*-expressing neurons in POA (Fig. S3a, c), but undetectable *Kiss1* expression in the ARC of male and female mice, and in the AVPV of females; areas where *Kiss1*-expressing neurons were abundantly detected in control animals (Fig. S3b, d). On the contrary, GoDKO mice showed undetectable levels of *GnRH* expression vs. controls (Fig. S3e, g), whereas the expression of *Kiss1* was not only preserved but even enhanced in the ARC of male and female mice, due to suppression of negative feedback control caused by the hypogonadal state of GoDKO animals (Fig. S3f). In turn, *Kiss1* expression in the AVPV of GoDKO females was markedly reduced (Fig. S3h), as ovarian steroids are known to enhance *Kiss1* mRNA levels selectively at this site[27].

In terms of neuroendocrine responses, KiDKO and GoDKO mice presented preserved responses to effective doses of GnRH, whose primary target is the pituitary, with significant increases in serum LH levels. Admittedly, the magnitude of these responses was smaller than in controls, but this is probably due to the diminished priming activity of endogenous GnRH at the pituitary of these animals rather than a primary major pituitary defect. In contrast, the expected rise of circulating LH following removal of negative feedback of gonadal secretions by gonadectomy did not occur in any of the two genotypes (Fig. S4a, b); similarly, no post-castration rise of FSH was detected in either KiDKO or GoDKO mice. However, disparate patterns of LH responses to kisspeptin-10 (Kp-10) and the glutamate receptor agonist, NMDA, were found between KiDKO and GoDKO mice. Thus, while KiDKO mice retained their capacity to respond to Kp-10 and NMDA (Fig. S4a), albeit with modest increases in LH secretion after NMDA, GoDKO mice displayed a complete ablation of LH responses to Kp-10 and NMDA (Fig. S4b).

## Late-onset HH in KiDKO mice: sexually different impact on puberty onset and attainment of fertility

The presence of a complete HH phenotype in mature adult KiDKO mice prompted us to explore earlier maturational (infantile, peripubertal) stages and compare with features in the GoDKO model. Male and female KiDKO mice displayed normal body weight gain during the infantile-juvenile period, as well as fully preserved phenotypic signs of pubertal onset, namely balano-preputial separation (BPS) in males and vaginal opening (VO) in females (Fig. S5a, c). Likewise, LH and FSH levels were similar in control and KiDKO mice of both sexes, at the infantile (2-week-old) and peripubertal (4-week-old) periods, except for a slight elevation of LH levels in KiDKO female mice at 4 weeks (Fig. 2a, c). In the same vein, no differences were detected in serum sex steroid levels between peripubertal KiDKO and control mice (Table 1). In contrast, GoDKO mice of both sexes displayed a profound hypogonadotropic state, with significantly lower LH and FSH levels (Fig. 2b, d), at 4 weeks of age, which was preceded by trends of reduced LH and FSH secretion already at 2 weeks, which reached statistical significance for FSH in 2-week-old GoDKO females (Fig. 2d). Yet, no significant changes were detected in circulating sex steroid levels in peripubertal GoDKO mice, except for a significant reduction in serum progesterone levels in males (Table 1), although estradiol concentrations were in the limit of detection also in control mice at this age. Body weight gain was preserved in GoDKO mice of both sexes during the prepubertal transition (Fig. S5b, d), with normal VO in pubertal GoDKO females (Fig. S5d). In contrast, GoDKO males did not display BPS (Fig. S5b), linked to incomplete penis development (micro-penis).

Testis weight gain during the prepubertal transition was not grossly affected in KiDKO male mice (Fig. 2e), although seminiferous tubule diameter was reduced at 4 weeks of age (Fig. 2f), which was associated with a lower percentage of elongated spermatids (Fig. S5e), without increased number of apoptotic germ cells (Fig. S5f). In turn, GoDKO males showed significantly decreased testis weight at 4 weeks of age (Fig. 2g), when the tubular diameter was also suppressed (Fig. 2h). In addition, the number of elongated spermatids was reduced and the percentage of tubules with apoptotic germ cells tended to be higher in GoDKO males than in controls, although the latter difference did not reach statistical significance (Fig. S5g, h).

In female KiDKO mice, ovarian weight gain during the prepubertal transition was not affected; yet, the uterus weight in 4-week-old KiDKO

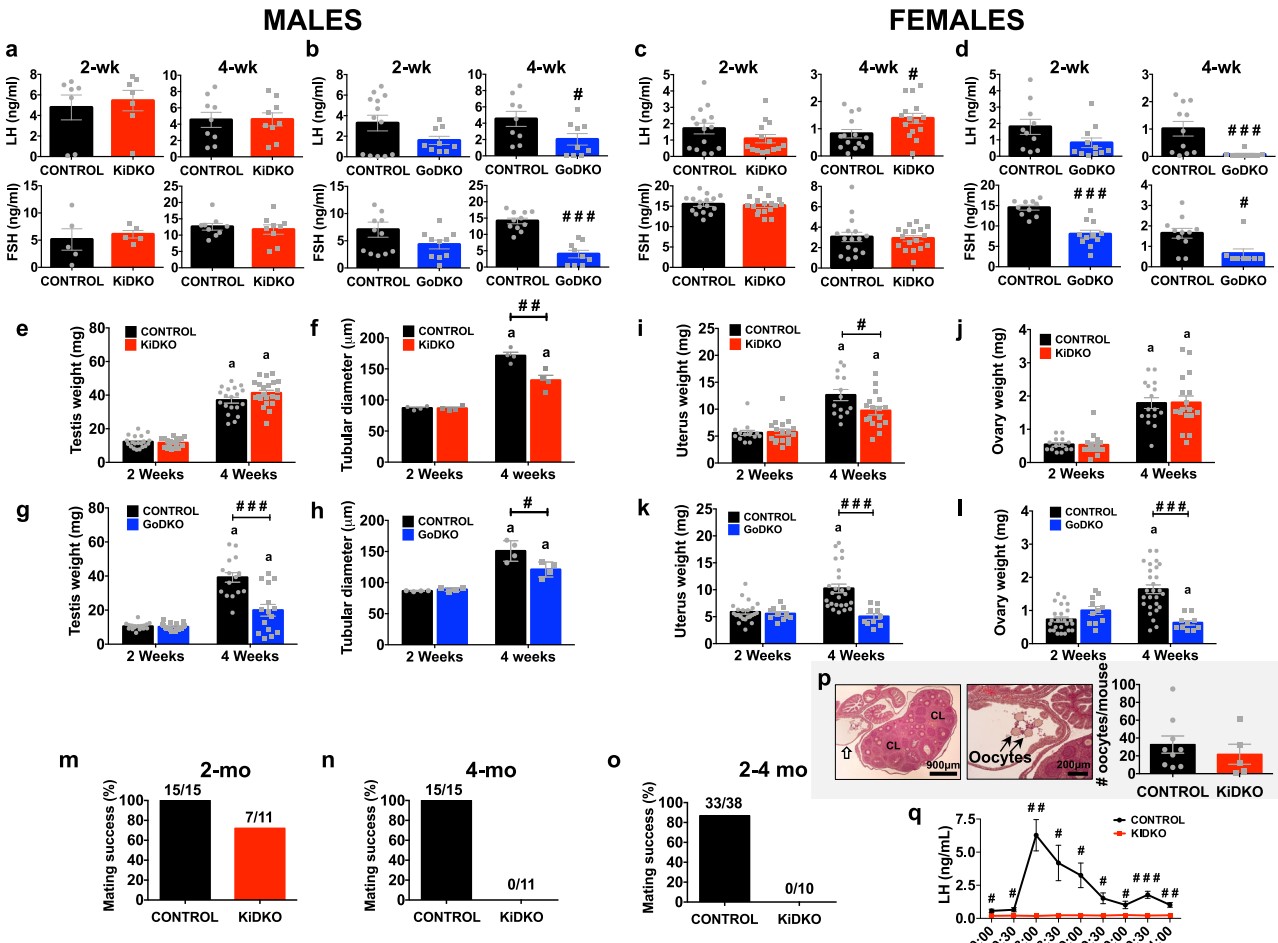

**Fig. 2 | Reproductive indices in KiDKO and GoDKO mice during the infantile-pubertal transition.** Data from the two models, KiDKO (Kiss1-specific Dicer KO) and GoDKO (GnRH-specific Dicer KO), are presented. Serum LH and FSH levels in KiDKO (**a**; LH in males: control $n = 7$, KiDKO $n = 7$ for 2 week and control $n = 9$, KiDKO $n = 9$ for 4 week; FSH in males: control $n = 5$, KiDKO $n = 5$ for 2 week and control $n = 9$, KiDKO $n = 9$ for 4 week; **c**; LH in females: control $n = 15$, KiDKO $n = 14$ for 2 week and control $n = 14$, KiDKO $n = 15$ for 4 week; FSH in females: control $n = 16$, KiDKO $n = 17$ for both 2 and 4 week) and GoDKO (**b**; LH in males: control $n = 13$, GoDKO $n = 9$ for 2 week and control $n = 9$, GoDKO $n = 9$ for 4 week; FSH in males: control $n = 12$, GoDKO $n = 10$ for 2 week and control $n = 11$, GoDKO $n = 9$ for 4 week; **d**; LH in females: control $n = 10$, GoDKO $n = 11$ for 2 week and control $n = 11$, GoDKO $n = 9$ for 4 week; FSH in females: control $n = 10$, GoDKO $n = 11$ for 2 week and control $n = 11$, GoDKO $n = 8$ for 4 week) mice, at 2 and 4 weeks of age, are shown. In addition, testicular weight and tubular diameter in KiDKO males (**e**; control $n = 20$, KiDKO $n = 18$ for 2 week and control $n = 19$, KiDKO $n = 22$ for 4 week; **f**; control $n = 4$, KiDKO $n = 4$ for both 2 and 4 week) and GoDKO (**g**; control $n = 18$, GoDKO $n = 19$ for 2 week and control $n = 16$, GoDKO $n = 15$ for 4 week; **h**; control $n = 4$, GoDKO $n = 4$ for both 2 and 4 week), as well as uterus and ovarian weights in KiDKO (**i**; control $n = 14$, KiDKO $n = 17$ for both 2 week and 4 week; **j**; control $n = 14$, KiDKO $n = 16$ for 2 week and control $n = 16$, KiDKO $n = 17$ for 4 week) and GoDKO (**k**; control $n = 23$, GoDKO $n = 11$ for 2 week and control $n = 25$, GoDKO $n = 10$ for 4 week; **l**; control $n = 24$, GoDKO $n = 11$ for 2 week and control $n = 27$, GoDKO $n = 10$ for 4 week) females are shown. Fertility tests were conducted in male and female KiDKO mice; results from tests carried out in KiDKO males at 2 months (**m**) and 4 months (**n**) of age, and in KiDKO females of 2–4 months (aggregated; see **o**) are presented. In **p**, histological images of the ovary of KiDKO females after ovulation induction by standard gonadotropin priming are shown (control $n = 9$, KiDKO $n = 5$), with appearance of newly formed corpora lutea (CL) and released oocytes in the uterine tubes (denoted by arrows). Quantification of the #oocytes observed in control and KiDKO females after gonadotropin priming is also displayed. Finally, control and KiDKO females were challenged with a protocol for sex steroid-induced LH surge (**q**; control $n = 5$, KiDKO $n = 5$). The values are represented as the mean ± SEM. #$P < 0.05$; ##$P < 0.01$; ###$P < 0.001$ vs. corresponding control groups; ª$P < 0.05$ vs. 2-week-old groups.

females was significantly suppressed (Fig. 2i, j). Anyhow, no morphometric alterations were detected in the postnatal ovary of KiDKO animals, with preserved numbers of resting, primary, secondary and antral follicles at 2- and 4 weeks of age and similar percentages of atretic follicles to control mice (Fig. S5i, j). Despite such features of preserved follicular maturation during the prepubertal transition, pubertal KiDKO females did not reach an ovulatory stage, as evidenced by the lack of first estrus and absence of corpora lutea (see also Fig. 1g, h). In GoDKO female mice, ovarian and uterus weights were preserved at 2 weeks of age, but these were significantly suppressed in 4-week-old females (Fig. 2k, l), so that the increase of uterus and ovarian weights, observed during the infantile-pubertal transition in control and KiDKO mice, was abolished in GoDKO females (Fig. 2i, l). In

addition, although no differences were found in the number of resting and growing follicles during the infantile and peripubertal period, the percentage of atretic follicles was significantly higher in the ovaries of GoDKO mice at 4 weeks of age (Fig. S5k, l), and they failed also to reach an ovulatory stage.

Finally, fertility tests were conducted in male and female mice of both genotypes. At two months of age, KiDKO males were proven fertile, with 7 out 11 virgin control females being impregnated (Fig. 2m). However, none of the KiDKO males were fertile at 4 months of age (0 out 11), while all of control males produced offspring when mated with virgin females, at both ages (Fig. 2n). Conversely, in line with signs of lack of spontaneous first ovulation, KiDKO females were infertile at 2–4 months of age, as shown by the lack of any successful pregnancy

after matting with male mice of proven fertility (Fig. 2o). Interestingly, however, prepubertal KiDKO females could be primed to ovulate, with ovulatory responses that were grossly similar to those of controls after a standard priming protocol with gonadotropins (Fig. 2p). Yet, KiDKO females failed to display LH surges in response to an effective protocol of sex steroid priming in ovariectomized mice (Fig. 2q). On the other hand, both male and female GoDKO mice were infertile, but prepubertal GoDKO females could be forced to ovulate after gonadotropin priming, in line with our previous data[25]; yet, the number of released oocytes was significantly lower than in control mice (Fig. S5m).

## Differential impact of Dicer ablation on Kiss1 expression in ARC and AVPV along postnatal maturation

The neuronal basis for the reproductive phenotype of KiDKO mice during postnatal maturation was explored by expression analyses conducted in control and conditional null animals of both sexes, at 2 weeks and 4 weeks of age, and in adulthood. In situ hybridization allowed detection of numbers of Kiss1-expressing neurons in the ARC (in both males and females) and AVPV (only in females), together with relative Kiss1 expression per neuron (Fig. S6). In parallel, we took advantage of the fact that our Kiss1-Cre line express also GFP (tagged to Cre protein) under the endogenous Kiss1 promoter to detect the presence of GFP-positive neurons, as marker of viable Kiss1 neurons, in KiDKO and control mice (Fig. 3); since CreGFP is a non-secreted protein, GFP labeling is expected to be more robust than Kiss1/kisspeptin, therefore increasing the window of detection of Kiss1-expressing neurons.

Kiss1 neuronal populations, denoted by GFP labeling, were fully preserved in the ARC of KiDKO mice of both sexes, and the AVPV of females, during the infantile period (2 weeks of age; Fig. 3a, d, g), with conserved numbers of Kiss1-expressing cells (Fig. S6a, d, g), except for a moderate decline detected in the ARC of infantile KiDKO males (Fig. S6a). Yet, Kiss1 mRNA expression per neuron was fully conserved (Fig. S6a; inset). In contrast, the number of Kiss1 neurons (Fig. 3b, h), Kiss1-expressing cells, and relative Kiss1 mRNA levels per neuron (Fig. S6b, h) in the ARC were significantly suppressed in 4-week-old male and female KiDKO mice. However, none of these parameters was affected in the AVPV of pubertal KiDKO females (Fig. 3e and Fig. S6e). Finally, the number of GFP-labeled Kiss1 neurons (Fig. 3c, f, i), and Kiss1 expression (Fig. S6c, f, i), dropped to (nearly) undetectable levels in the ARC and AVPV populations of adult KiDKO of both sexes.

## Differential impact of Dicer ablation on Kisspeptin levels in ARC and AVPV along postnatal maturation

Given the proven role of miRNAs in the post-transcriptional control of gene expression, kisspeptin protein levels were assessed in KiDKO mice at similar ages and at hypothalamic sites. Kiss1-specific Dicer ablation resulted in a massive drop in kisspeptin-immunoreactivity in the ARC of both males and females, which was observed already at 2 weeks of age, when protein levels were already nearly undetectable (Fig. 4a, b, g, h). In contrast, no significant differences in kisspeptin content/kisspeptin-positive cells were detected in the AVPV of KiDKO and control mice at 2- and 4-weeks of age (Fig. 4d, e). Of note, despite the massive suppression of kisspeptin content in the ARC, the infantile-pubertal transition was associated with an increase in kisspeptin-immunoreactivity at this nucleus in KiDKO mice of both sexes, which was accompanied by a significant rise in the number of kisspeptin-positive cells also in the AVPV (Fig. S7). Finally, kisspeptin-immunoreactivity was massively suppressed in adult KiDKO mice of both sexes, not only at the ARC (Fig. 4c, i) but also at the AVPV (Fig. 4f).

## Differential impact of Dicer ablation on NKB vs. Kisspeptin expression in ARC KNDy neurons

NKB immunoreactivity in the ARC was also affected by Dicer ablation in Kiss1-expressing cells. Yet, the profiles of protein content clearly differed from those of kisspeptin. Thus, while kisspeptin-immunoreactivity was markedly suppressed in the ARC of KiDKO mice of both sexes at the infantile period, no statistical differences in the density of NKB fibers were detected in the ARC between KiDKO and control male and female mice, at this age (2 weeks; Fig. 5a, d). In contrast, NKB protein levels were significantly diminished in the ARC of KiDKO males and females at the peripubertal period (4 weeks; Fig. 5b, e). However, the relative magnitude of such suppression was substantially milder than that of kisspeptin. Thus, while ARC kisspeptin-immunoreactivity was reduced by >160- and 10-fold in pubertal male and female KiDKO mice, respectively, the decrease in NKB protein levels was only five- and two-fold in KiDKO males and females, respectively (Fig. 5b, e). In addition, while NKB-immunoreactivity augmented during the infantile-pubertal transition in KiDKO males, this effect was not detected in conditional null females (Fig. S7). Finally, the drop in NKB immunoreactivity was maximal in adult KiDKO mice of both sexes, in which only a few NKB fibers were detected (Fig. 5c, f). This decrease in the level of NKB protein content appears to be specific to KNDy neurons, since NKB immunoreactivity was preserved in other brain regions (Fig. S8), known to harbor NKB neurons[28].

## Impact of Dicer ablation on Kiss1 neuronal survival

To track Kiss1 neuronal survival following congenital Dicer ablation, a triple transgenic mouse line, expressing the Cre-dependent reporter (ROSA26)-YFP in Kiss1 neurons was generated on the KiDKO background. This approach permits persistent labeling of any cell ever expressing Kiss1, even if Kiss1 expression is no longer active. Analyses in this reporter mouse line were conducted at 4 weeks, 2 months, and 4 months of age (Fig. 6); namely, the developmental window in which kisspeptin (and NKB) content substantially decrease. Kiss1-YFP neurons in the ARC of KiDKO male mice were diminished already at the peripubertal period (4 weeks; Fig. 6a); yet, such a decline was substantially milder than that of kisspeptin-immunoreactivity and Kiss1 mRNA expression, which dropped to nearly negligible levels at this age (see Fig. 4b), therefore suggesting that such a decline in neuronal survival is not driven by the loss of kisspeptin per se. The decrease in the number of Kiss1-YFP cells progressed during the adult stage, with a substantial reduction in 4-month-old KiDKO males (Fig. 6b, c), which nonetheless was less pronounced than the suppression of kisspeptin content and Kiss1 expression.

In KiDKO female mice, the number of Kiss1-YFP cells in the ARC of peripubertal (4-week-old) animals was fully preserved (Fig. 6g), despite a marked reduction of kisspeptin and NKB content and Kiss1 expression, at this age. In turn, the number of Kiss1-YFP neurons was diminished in the ARC of adult KiDKO females at 2- and 4-months of age (Fig. 6h, i). However, KiDKO females retained a substantial number of Kiss1-YFP neurons at the ARC, both at 2 months (1110 ± 116) and 4 months (579 ± 45) of age, which is in contrast with the massive suppression of kisspeptin and NKB content, and Kiss1 levels, in the ARC of adult KiDKO females, in which the expression of these neuropeptides was almost null. Notably, the number of Kiss1-YFP was totally conserved in the AVPV of KiDKO female mice, at least up to the age of 2 months, it being reduced only in 4-month-old animals (Fig. 6d–f).

## Dicer ablation induces upregulation of repressor expression in ARC Kiss1 neurons

To assess the molecular mechanisms involved in the disruption of Kiss1 neuronal function and Kiss1 expression after conditional elimination of Dicer in Kiss1 cells, we performed expression analyses of different regulators of Kiss1 promoter activity in Kiss1 neurons isolated by FACS from the ARC and AVPV of peripubertal (PND28) control and KiDKO mice of both sexes. To this end, we selected a number of genes previously identified as repressors of the Kiss1 promoter, i.e., Mkrn3, Sirt1, Eap1, Cux1, and the Polycomb group (PcG) members, Eed Cbx7 and Yy1, with demonstrated inhibitory actions on pubertal progression[14,26,29–31].

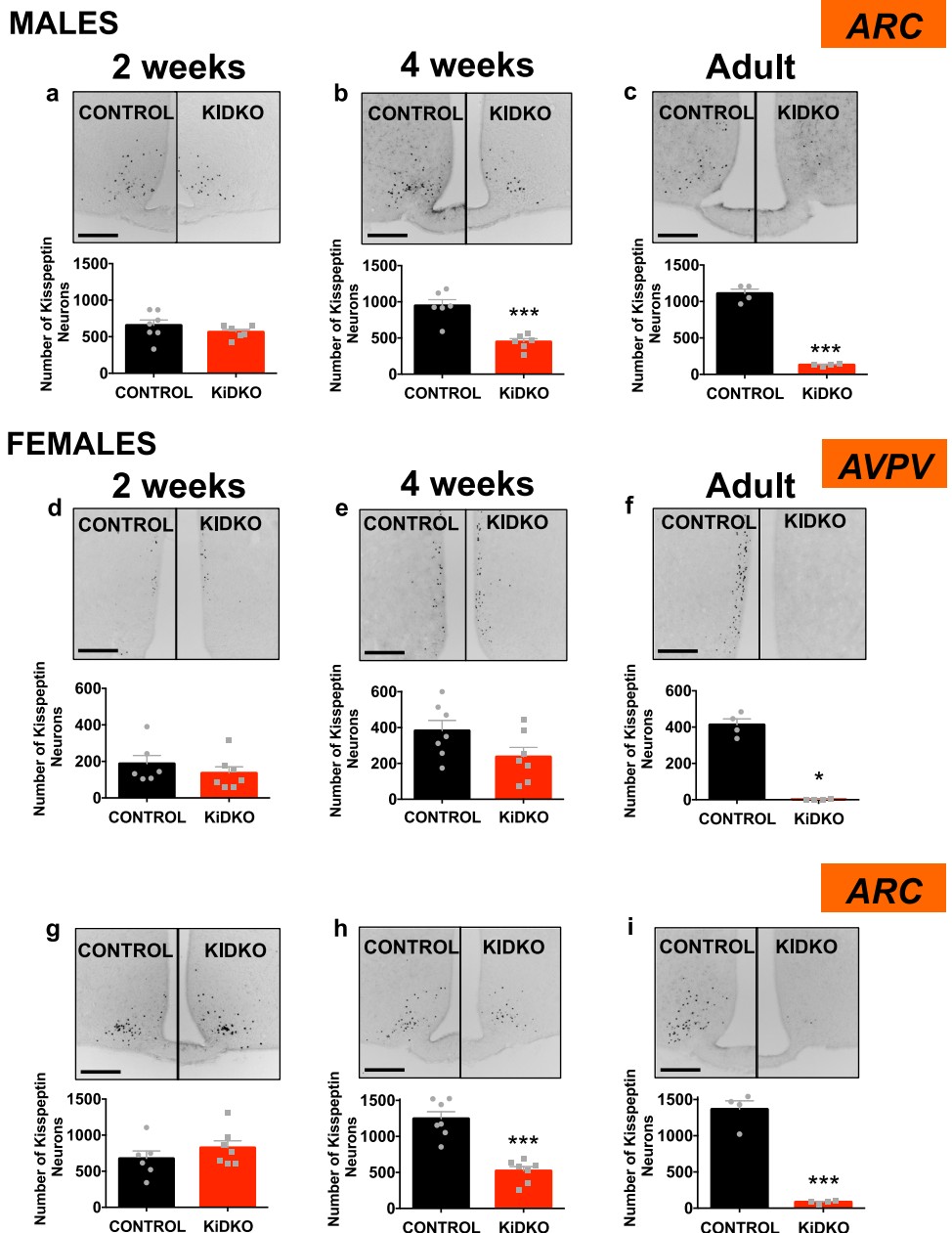

**Fig. 3 | Number of GFP-labeled, Kiss1-expressing neurons in KiDKO mice along postnatal maturation.** Taking advantage of expression of GFP (tagged to Cre) in Kiss1 neurons of our KiDKO line, GFP-positive (+ve) cells were counted, assuming that, since CreGFP is a non-secreted protein, GFP labeling would provide a more robust maker of viable Kiss1-expressing neurons. Representative images and quantitative data on GFP+ve cells in the ARC of males (**a**–**c**), and the AVPV (**d**–**f**), and ARC (**g**–**i**) of females, of control and KiDKO genotypes, are presented. Data were collected at three postnatal ages: 2 weeks (corresponding to mini-puberty), 4 weeks (corresponding to early pubertal transition), and adulthood (4 months). Group sizes: ARC: $n = 7$, 6, and 4 for 2 weeks, 4 weeks, and adult, control males, respectively; $n = 6$, 6, and 4 for 2 weeks, 4 weeks, and adult KiDKO males, respectively; $n = 6$, 7, and 4 for 2 week, 4 week, and adult control females, respectively; $n = 7$, 7, and 4 for 2 weeks, 4 weeks, and adult of KiDKO females, respectively; AVPV: $n = 6$, 7, and 4 for 2 weeks, 4 weeks, and adult control females, respectively; $n = 6$, 7, and 4 for 2 weeks, 4 weeks, and adult KiDKO females. The values are represented as the mean ± SEM. *$P < 0.05$; ***$P < 0.001$ vs. corresponding control groups. Scale bars correspond to 200 μm.

For comparative purposes, we measured also the expression levels of two reported *Kiss1* transcriptional activators, Ttf1 and the Trithorax group member, Mll1[29,32].

Quantitative RT-PCR analyses demonstrated a significant increase in the expression of *Kiss1* transcriptional repressors, *Mkn3*, *Cbx7*, and *Eap1*, in ARC Kiss1 neurons from both male and female KiDKO mice (Fig. 7a), which is compatible with the decrease of *Kiss1* expression detected during this time-window (Fig. 3b, h and Fig. S6b, h). Interestingly, this effect was not detected in AVPV Kiss1 neurons (Fig. 7c), where *Kiss1* expression was not affected at this age point (Fig. 3e and

Fig. S6e). Moreover, *Yy1* expression was modestly, but significantly reduced in AVPV Kiss1 neurons from KiDKO females (Fig. 7c). On the other hand, expression of the *Kiss1* transcriptional activator, *Mll1*, tended to be (non-significantly) increased in ARC, but not AVPV, Kiss1 neurons of KiDKO animals, while no changes were detected in the expression of the other activator, *Ttf1*, in either ARC or AVPV Kiss1 neurons (Fig. 7b, d).

A graphical summary of the major neuroendocrine and phenotypic alterations induced by conditional ablation of Dicer in Kiss1 cells is shown in Fig. 8.

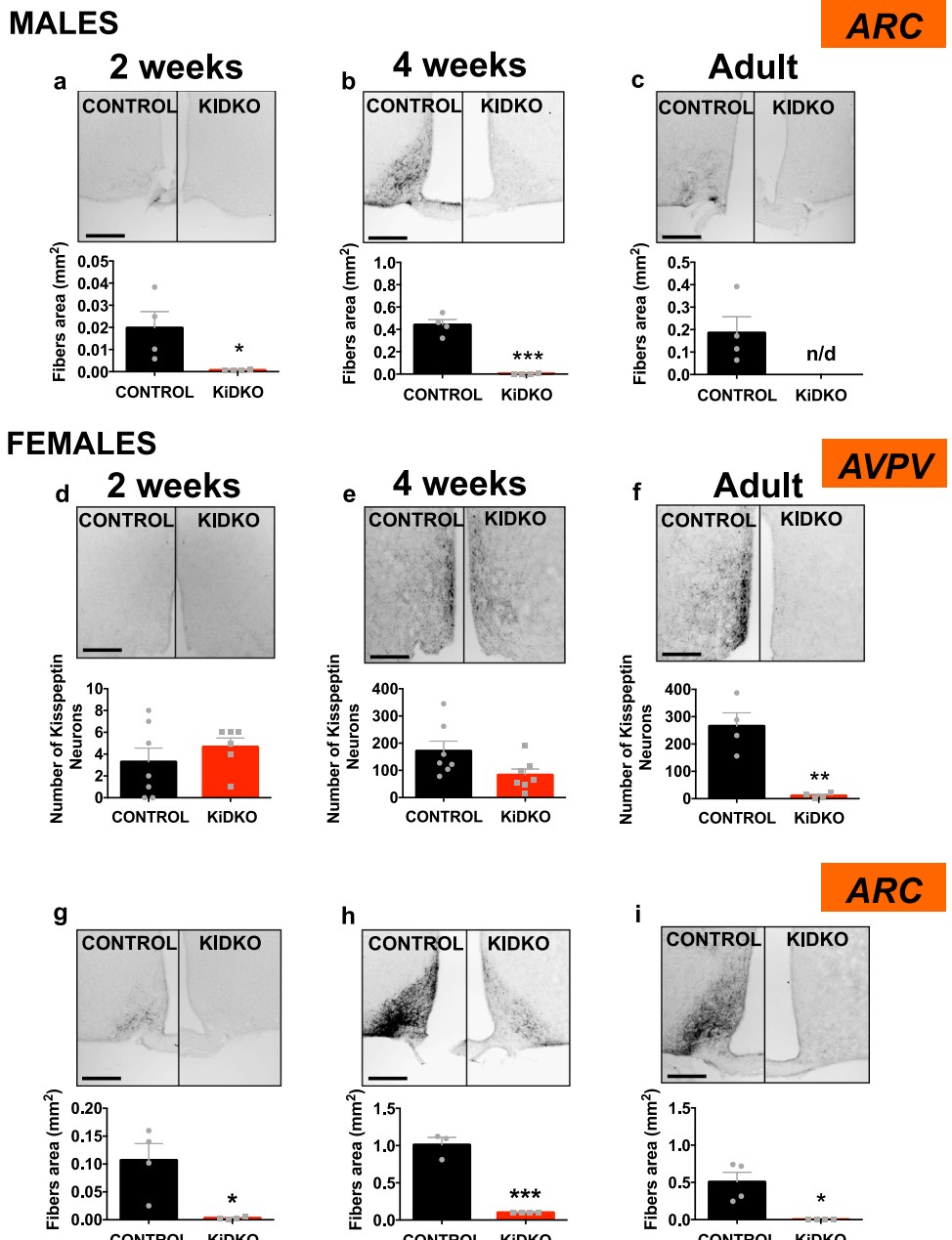

**Fig. 4 | Hypothalamic kisspeptin-immunoreactivity in KiDKO mice along postnatal maturation.** Detection of kisspeptin content in situ was conducted using immunohistochemistry. Due to the features of immunoreactivity (IR) of kisspeptin in the mouse hypothalamus, this procedure allowed the detection of fibers in the ARC, while permitting the counting of numbers of kisspeptin-IR cells in AVPV. Representative images and quantitative data on kisspeptin-IR fibers (area) or cells in the ARC of males (**a**–**c**) and the AVPV (**d**–**f**) and ARC (**g**–**i**) of females of control and KiDKO genotypes are presented. Data were collected at three postnatal ages: 2 weeks (corresponding to mini-puberty), 4 weeks (corresponding to early pubertal transition), and adulthood (4 months). Group sizes for ARC densitometry: $n = 4$ control males, $n = 4$ KiDKO males for 2 week, 4 week, and adults; $n = 4$ control females; $n = 4$ KiDKO females for 2 week and adults; and $n = 3$ control females; $n = 4$ KiDKO females for 4 week. Of note, for quantification of kisspeptin cell numbers in the AVPV, higher number of animals at 2- and 4-weeks was included due to the greater variability: $n = 7$ control females, $n = 6$ KiDKO females for 2 weeks, and $n = 7$ control females, $n = 7$ KiDKO females for 4 week. The values are represented as the mean ± SEM. *$P < 0.05$; **$P < 0.01$; ***$P < 0.001$ vs. corresponding control groups. n/d not detectable. Scale bars correspond to 200 μm.

## Discussion

In the last decades, miRNAs have emerged as key regulatory elements of a large number of biological processes, acting mainly as post-transcriptional repressors of target genes, via interfering protein translation or promoting RNA degradation[16,17]. We show here that congenital ablation of Dicer, the key enzyme for mature miRNA synthesis, in Kiss1 neurons caused a state of profound hypogonado-tropic hypogonadism in mature adult mice of both sexes, with complete infertility. This phenotype occurred despite the preserved expression of *GnRH* in the POA and conserved functionality of GnRH neurons, denoted by detectable LH responses to central kisspeptin administration in KiDKO mice, and was seemingly caused by almost complete elimination of *Kiss1* expression and kisspeptin content in the AVPV and ARC in adult animals. While fragmentary data have very recently suggested putative roles of specific miRNAs in the control of *Kiss1* in other cellular contexts, ranging from ectopic pregnancy[33]

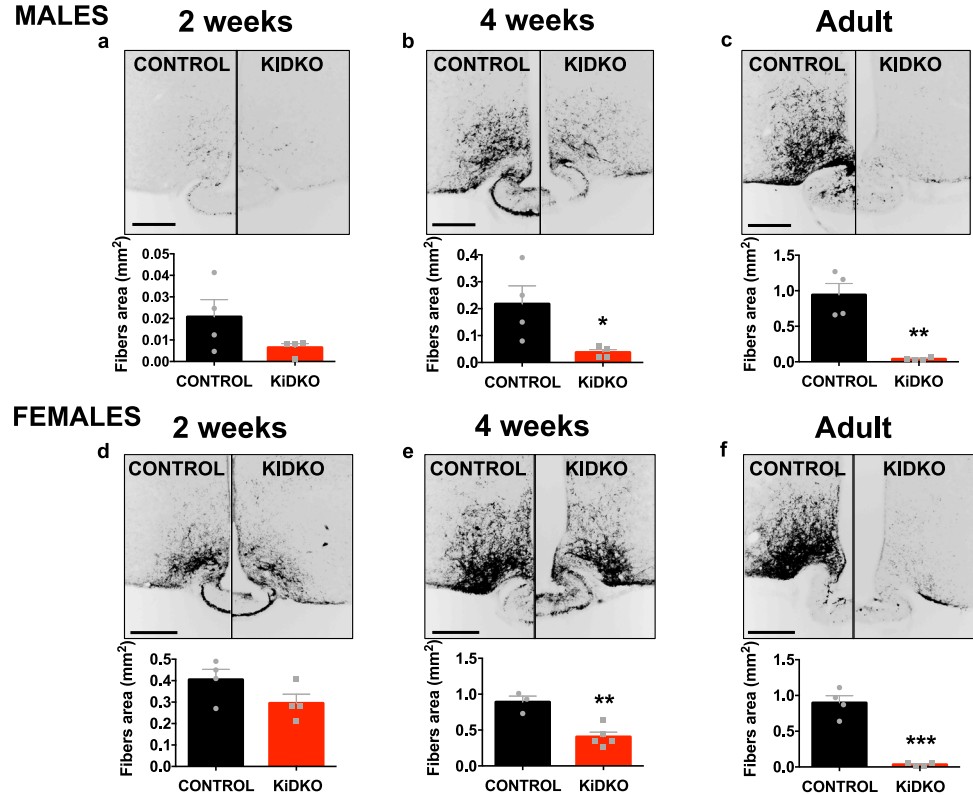

**Fig. 5 | Hypothalamic NKB immunoreactivity in KiDKO mice along postnatal maturation.** Detection of NKB content in situ was conducted using immunohistochemistry, which allowed labeling of fibers with NKB immunoreactivity (IR). Representative images and quantitative data on NKB-IR fibers (area) in the ARC of male (**a**–**c**) and female (**d**–**f**) KiDKO mice, and their corresponding controls, are presented. Data were collected at three postnatal ages: 2 weeks (corresponding to mini-puberty), 4 weeks (corresponding to early pubertal transition); and adulthood (4 months). Group sizes: $n = 4$ control males, $n = 4$ KiDKO males for 2 weeks, 4 weeks, and adults; $n = 4$ control females; $n = 4$ KiDKO females for 2 weeks and adults and $n = 3$ control females; $n = 5$ KiDKO females for 4 weeks. The values are represented as the mean ± SEM. *$P < 0.05$; **$P < 0.01$; ***$P < 0.001$ vs. corresponding control groups. Scale bars correspond to 200 μm.

to brain cancer metastasis[34] and immortalized GT1-7 cells[35], our data provide conclusive in vivo evidence for an essential role of miRNA biogenesis in Kiss1 neurons for proper functioning of the reproductive axis in adulthood, and further illustrate the indispensable role of kisspeptin input onto GnRH neurons for adult fertility. Interestingly, defective *Kiss1*/kisspeptin expression in our model of conditional ablation of Dicer was not primarily caused by an early massive loss of Kiss1 neurons, due to compromised survival linked to impairment of miRNA biosynthesis, since a significant proportion of these neurons remained present in young-adult KiDKO mice. In contrast, prevention of mature miRNA generation seemingly halted kisspeptin production in Kiss1 neurons.

In order to assess the mechanism underlying such molecular phenotype, we did not search for differentially-expressed miRNAs in our KiDKO mice, as we presumed mature miRNA biogenesis was largely blunted following congenital elimination of Dicer. In fact, despite evidence for non-canonical, Dicer-independent synthesis of certain miRNAs, such as miR-451, in vertebrates[36,37], the expected alteration of a very high number of miRNAs in Kiss1 neurons from KiDKO mice, due to the conditional ablation of the canonical pathway, would have prevented us from pinpointing specific miRNAs responsible for the phenotype. Likewise, we assumed that ablation of miRNAs from Kiss1 neurons is unlikely to cause repression of *Kiss1* expression directly, given the proposed role of miRNA as gene silencers[17]. In turn, we hypothesized that global suppression of the miRNA landscape in Kiss1 neurons might have lifted the expression of Kiss1 repressors, thereby inhibiting *Kiss1*, as has been previously proposed for miRNA regulation of GnRH and its suppression in GoDKO mice[25]. To test this possibility, expression analyses of a panel of seven repressors and two activators

of *Kiss1*, selected on the basis of previous studies[14,29–32], were conducted in FACS-isolated Kiss1 neurons, obtained from pubertal KiDKO mice of both sexes. In line with our working hypothesis, three reported repressors of *Kiss1*, namely *Mkrn3*[31], *Cbx7*[30], and *Eap1*[29], were upregulated in ARC Kiss1 cells congenitally devoid of mature miRNA biogenesis, strongly suggesting that this is a major mechanism for the observed suppression of *Kiss1* in our conditional null model.

Mkrn3 is a puberty-suppressing factor, which has been recently shown to be expressed and operate in ARC Kiss1 neurons to repress *Kiss1* expression[31]. In a recent study, we have independently shown that Mkrn3 has three highly conserved seed regions for members of the miR-30 family in its 3′-UTR and that miR-30 represses Mkrn3 to modulate pubertal timing[26]. Intriguingly, our bioinformatic analyses indicate that the nuclear isoform of Cbx7, a member of the Polycomb group (PcG) of silencers that has been shown to epigenetically repress *Kiss1* expression[30], is also a potential target of miR-30. Thus, it is tenable that elimination of miR-30 by congenital ablation of Dicer has a dominant role in the observed upregulation of *Mkrn3* and *Cbx7*, and thereby suppresses *Kiss1* expression in KiDKO mice. Admittedly, however, ablation of other miRNAs is likely to contribute to this molecular phenotype, as bioinformatic predictions failed to identify seed regions of miR-30 among the putative miRNA regulators of the 3′-UTR of *Eap1*. In any event, our data highlights a relevant miRNA-regulatory node, selectively targeting key transcriptional repressors, that operates in ARC Kiss1 neurons to precisely control *Kiss1* expression and is essential for reproductive function. This system seems to be functional in both sexes, but apparently not in AVPV Kiss1 cells, at least at the age-window analyzed, in line with the divergent impact of Dicer ablation in ARC vs. AVPV Kiss1 neurons. Collectively, our data

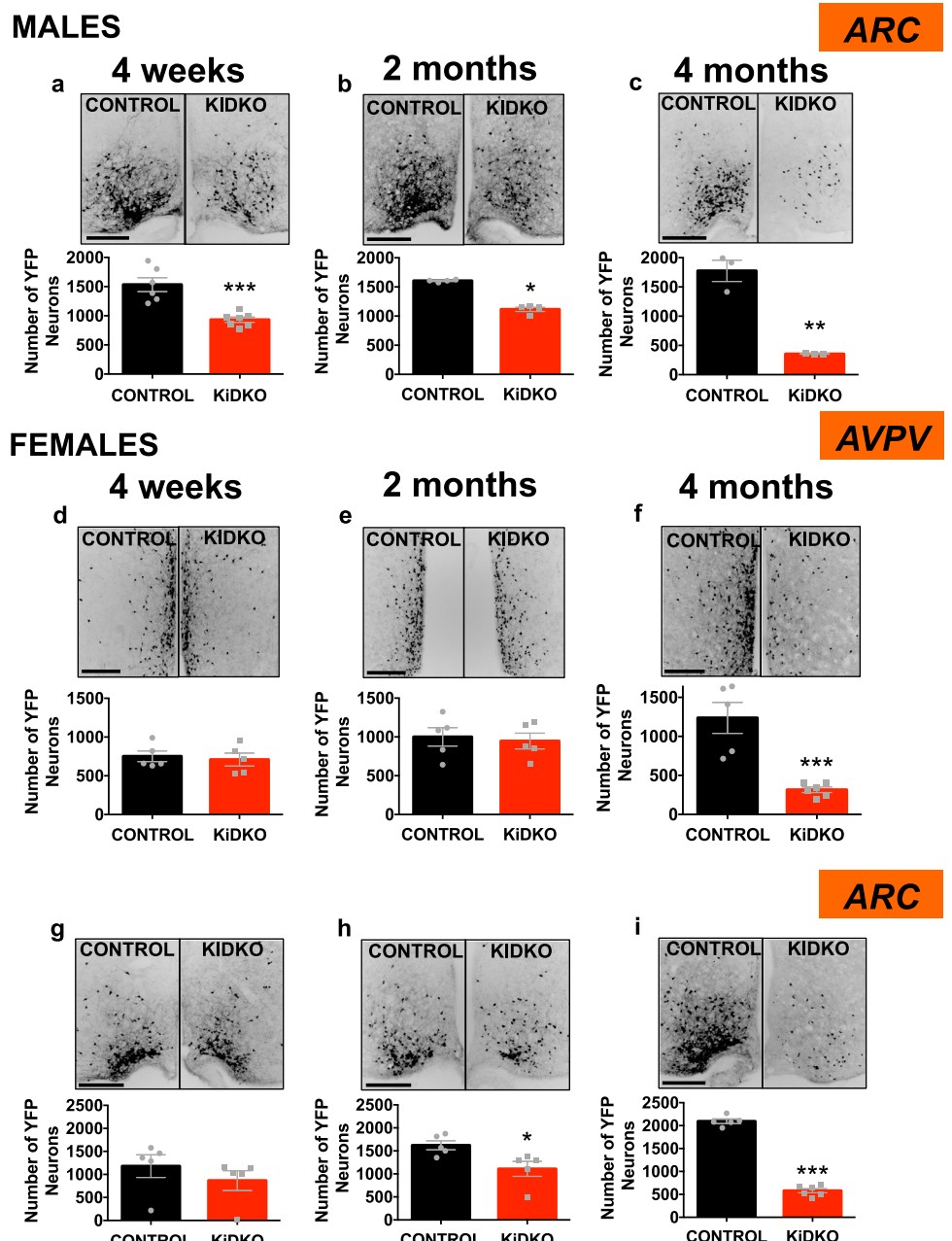

**Fig. 6 | Kiss1 neuronal survival in the hypothalamus of KiDKO mice along postnatal maturation.** Labeling of viable Kiss1 neurons in vivo was achieved by using a triple transgenic mouse line, expressing the Cre-dependent reporter (ROSA26)-YFP in Kiss1 neurons upon a KiDKO background. In this mouse line, cells ever expressing Kiss1 become persistently labeled with the fluorescent marker YFP, even if they stop expressing Kiss1. Representative images and quantitative data on YFP-positive cells in the ARC of males (**a**–**c**) and the AVPV (**d**–**f**) and ARC (**g**–**i**) of females of control and KiDKO genotypes are presented. Data were collected at three postnatal ages: 4 weeks (corresponding to early pubertal transition); and two periods of adulthood (2- and 4-months). Group sizes: $n = 6$, 4, and 3 for 4 weeks, 2 months, and 4 months, control males, respectively; $n = 7$, 4, and 3 for 4 weeks, 2 months, and 4 months KiDKO males, respectively; $n = 5$ for all control females; $n = 5$, 5, and 6 for 4 weeks, 2 months, and 4 months KiDKO females, respectively. The values are represented as the mean ± SEM. *$P < 0.05$; **$P < 0.01$; ***$P < 0.001$ vs. corresponding control groups. Scale bars correspond to 200 μm.

substantiate the physiological relevance of a system of "repressors-of-repressors" in Kiss1 neurons, which seems to operate also, albeit with different players, in GnRH neurons[25], and plays a fundamental role in the precise control of reproductive function.

Our study, which included parallel assessment of key neuroendocrine features in models of congenital ablation of Dicer in Kiss1 or GnRH neurons, allowed us to delineate also the specific roles of these key factors in the control of the reproductive axis in vivo. Notably, KiDKO and GoDKO mice displayed selective, compartmentalized alteration of the Kiss1 and GnRH systems, respectively. Thus,

abolished ARC and AVPV *Kiss1* expression, but fully preserved *GnRH* expression was observed in KiDKO animals; such conserved expression is compatible with a predominant post-transcriptional control of GnRH neurons by kisspeptin[38]. In contrast, in GoDKO mice, *GnRH* expression was totally absent, but Kiss1 neurons retained their capacity to respond to sex steroid feedback, with enhanced *Kiss1* expression in the ARC, compatible with the hypogonadal state of GoDKO mice. Despite these differences, KiDKO and GoDKO mice displayed notable similarities in terms of phenotypic presentation in adulthood, with overt central hypogonadism in both models, which could be reversed

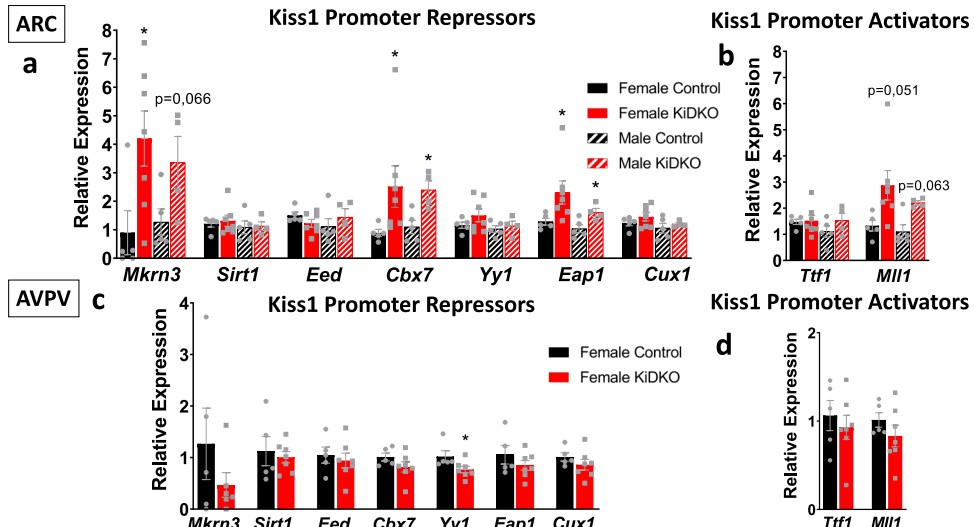

**Fig. 7 | Molecular mechanisms underlying Kiss1 neuronal alterations in KiDKO mice.** Relative expression of Kiss1 promoter repressors, Mkrn3, Sirt1, Eap1, Cux1, and members of the Polycomb group, Eed, Cbx7, and Yy1, were analyzed in ARC Kiss1 neurons isolated by FACS from control and KiDKO male and female mice (**a**) and in AVPV Kiss1 neurons from control and KiDKO female mice (**c**). In addition, Kiss1 promoter activators, Ttf1 and Mll1, were measured in the same ARC (**b**) and AVPV (**d**) Kiss1 neuronal samples. Groups sizes: $n = 5$ control females; $n = 7$ KiDKO females; $n = 5$ control males; $n = 4$ KiDKO males. The values are represented as the mean ± SEM. *$P < 0.05$ vs. corresponding control groups.

(in terms of ovulatory induction) by proper gonadotropin priming. This is compatible with a primary impact of Dicer ablation at central levels, while the putative effects of Dicer elimination in peripheral reproductive tissues, where modest *Kiss1* or *GnRH* expression has been reported previously[1], seem to negligibly contribute to this phenotype.

Anyhow, compatible with a more distal role of GnRH neurons as final output pathway for the brain control of the reproductive axis[2], GoDKO animals displayed signs of more profound suppression of gonadal (denoted by sex steroid levels) and gonadotropic function, as well as abolished gonadotropin responses to central activators, such as kisspeptin and NMDA, which were preserved in KiDKO mice. Notwithstanding, adult male mice with congenital ablation of Dicer in Kiss1 neurons not only presented diminished basal gonadotropin and sex steroid levels but also failed to show the expected gonadotropin rise after gonadectomy, therefore supporting the notion that functional Kiss1 neurons are indispensable for mediating the negative feedback effects of sex steroids and gonadotropin responses to their withdrawal. In the same vein, KiDKO females did not display preovulatory-like LH surges in response to proper sex steroid priming, suggesting a central defect of positive feedback mechanisms. In addition, KiDKO mice showed alterations in tonic LH secretory profiles, as a surrogate marker of pulsatile GnRH neuro-secretion, in line with recent evidence supporting a master role of Kiss1 neurons in the control of the GnRH pulse generator[4]. Interestingly, these alterations consisted in a significant decrease in the amplitude of LH peaks, whereas pulse frequency was not altered, suggesting the persistence of pacemaker mechanisms even in the presence of diminished or absent kisspeptin inputs. All in all, our data conclusively demonstrate that both Kiss1 and GnRH neurons are compulsory for the maintenance of reproductive function in adulthood since neither of these populations, separately, was sufficient to sustain reproduction, and further stress the essential role of central kisspeptin input in the control of adult GnRH neurons and reproductive function. Admittedly, analogous observations had been previously made in models of genetic inactivation of *Gpr54* or *Kiss1* genes[39,40]. Yet, in those previous models, congenital ablation of these factors resulted in early disruption of kisspeptin signaling, which caused maturational alterations and lack of pubertal activation prior to infertility. Our current models add an interesting chronological perspective to the analysis of the developmental roles of kisspeptin

signaling, as defined by the progressive suppression of *Kiss1* expression in our KiDKO model, which manifested as diminished levels along pubertal progression and became null at the adult stage. This is analogous to the timeline of GnRH suppression previously reported in our GoDKO model[25], and strongly suggests that early maturational defects are not seemingly major contributing factors for the adult hypogonadal state that was observed in KiDKO (or GoDKO) mice.

In search of the neuroendocrine substrate for the progressive central hypogonadism seen in KiDKO mice, we analyzed changes in Kiss1 neurons, *Kiss1* expression, and kisspeptin content in the ARC and AVPV of male and female mice during the infantile-pubertal transition. Due to the inherent features of the hypothalamic Kiss1 system in the mouse, *Kiss1*/kisspeptin analyses in the AVPV were restricted to females, whereas ARC expression analyses in both sexes included also assessment of NKB, as major co-transmitter of KNDy neurons at this hypothalamic site. Our analyses targeted two key maturational periods, namely mini-puberty (2 weeks of age) and the early pubertal transition (4 weeks of age). Notably, mini-puberty has been defined as a key developmental stage of initial activation of the gonadotropic axis, occurring both in humans and rodents, which is crucial for shaping later activation of reproductive function at puberty[41]. Indeed, recent evidence has suggested that major changes in miRNA regulation of GnRH neurons occur around mini-puberty in mice[25]. Our data conclusively document that congenital Kiss1-specific ablation of Dicer failed to diminish the number of Kiss1 neurons, either in the ARC or AVPV, of male and female mice at this period, with largely conserved *Kiss1* expression, except for a moderate (<30%) drop in infantile males, and totally conserved NKB content in the ARC at both sexes. In clear contrast, ARC kisspeptin content, as measured by immunohistochemistry, was markedly suppressed in male and female KiDKO mice at 2 weeks of age, while AVPV levels in females were not altered. Considering that indices of pubertal onset were grossly preserved in KiDKO animals of both sexes, these findings suggest that conservation of Kiss1 neuronal populations at mini-puberty, despite congenital Dicer ablation, is sufficient to activate the neuroendocrine pathways leading to puberty onset, even in the presence of substantially diminished kisspeptin content in the ARC. The later occurred in face of largely preserved *Kiss1* expression, which might be indicative of a compensatory response of enhanced secretion of kisspeptin from ARC

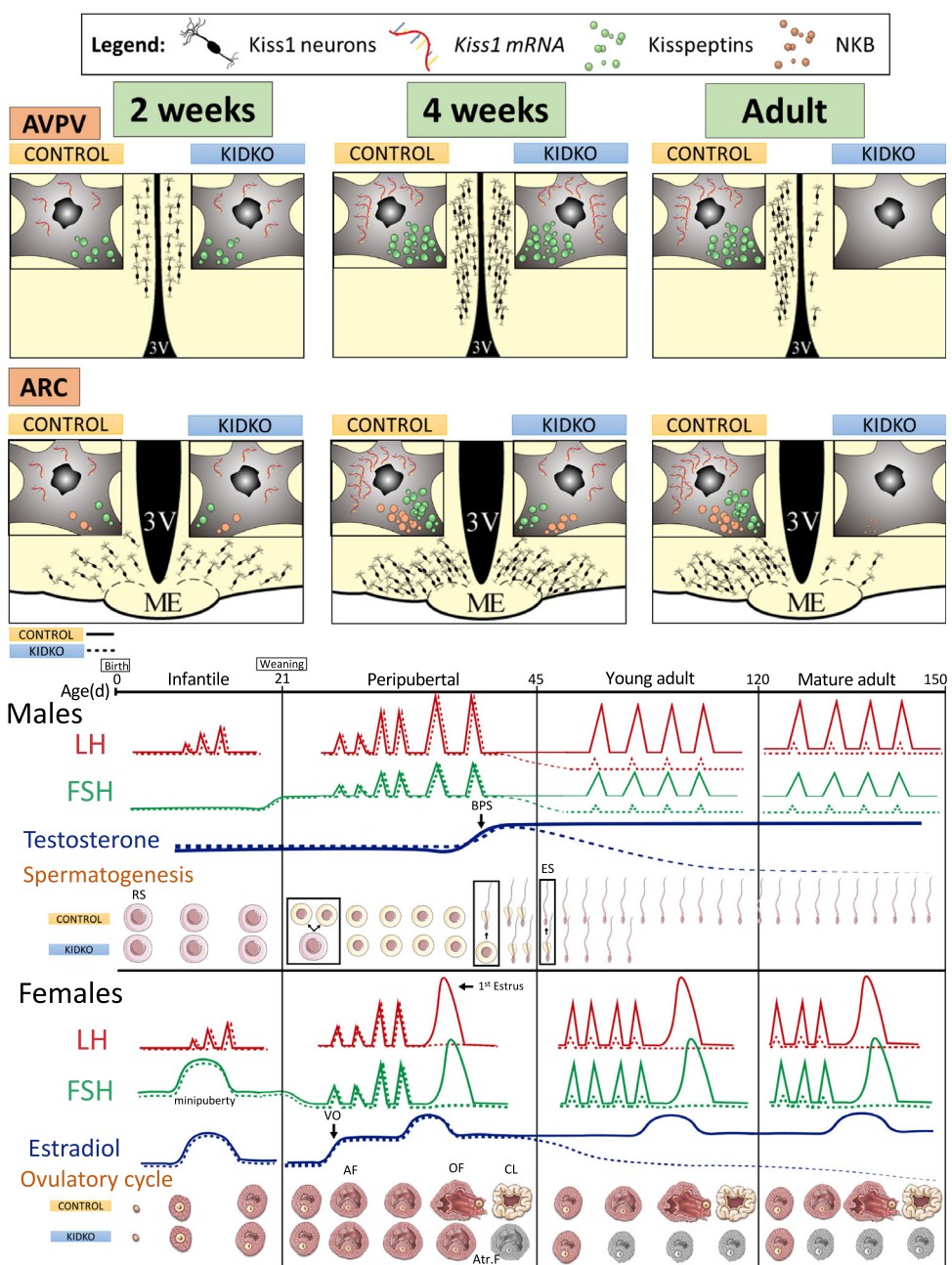

**Fig. 8 | Major neuroendocrine and phenotypic alterations after conditional ablation of Dicer in Kiss1.** A graphical summary of the major changes occurring in KiDKO mice is presented. In addition to changes in the number/survival of Kiss1 neurons, as well as expression of *Kiss1*, kisspeptin and NKB, in the AVPV and ARC of KiDKO vs. control mice, a recapitulation of major hormonal and reproductive (gonadal) alterations was observed in KiDKO animals is provided for both males and females. RS round spermatid, ES elongated spermatid, AF antral follicle, OV ovulatory follicle, Atr.F atretic follicle, CL corpus luteum.

Kiss1 neurons, which is compatible with the early signs of pubertal onset. In any event, the fact that the AVPV kisspeptin content, as well as ARC NKB levels, were fully preserved in infantile KiDKO mice may contribute also to preserved pubertal activation, and further stress that, rather than an unspecific impact on Kiss1 survival or function, ablation of mature miRNA biosynthesis evokes quite distinct functional alterations of Kiss1 neurons, with a massive depletion (or release) of kisspeptin being detected only in the ARC of both sexes.

Similar analyses at early stages of the pubertal transition revealed a sex-biased progressive impact of Dicer ablation on Kiss1 neurons and the attainment of fertility. Thus, while the total numbers of Kiss1 neurons were grossly preserved in the ARC of KiDKO female mice, they drop to nearly 60% of control values in conditional null males.

Likewise, *Kiss1* expression and NKB content was more severely suppressed in the ARC of KiDKO male mice, with values ranging 20−35% of control levels, while in females, expression levels were only half of the control values. In any event, kisspeptin content in the ARC was almost negligible in both male and female conditional null mice. In clear contrast, neither *Kiss1* expression nor kisspeptin content were significantly altered in the AVPV of female KiDKO mice, further emphasizing the differential impact of Dicer ablation between these two hypothalamic nuclei. Strikingly, while the effect of congenital elimination of Dicer on ARC Kiss1 neurons was more dramatic in males, attainment of fertility was detected in a majority of young (2-month-old) KiDKO male mice, while none of the females ever became fertile. These observations further substantiate the previous contention,

based on models of congenital ablation, that male puberty is less sensitive to suppression of Kiss1, as denoted by the fact that preservation of just a marginal fraction (~5%) of Kiss1 expression appeared sufficient for the achievement of male fertility[42], although compensatory mechanisms, of as yet unknown nature, were alluded to play a role in such phenomenon. Our current data, based on a model of progressive loss of Kiss1 expression, refine those previous findings, suggesting that preservation of a minute part of kisspeptin/NKB input during the pubertal transition is sufficient to complete puberty and attain fertility in the male, with development of an infertile phenotype later in adulthood, along with a more profound perturbation of the Kiss1 system with age. In clear contrast, completion of puberty and attainment of fertility in the female requires more robust preservation of the hypothalamic Kiss1 system, as denoted by the lack of first ovulation despite a less severe impact of Dicer ablation on ARC Kiss1 neurons and the apparently preserved AVPV Kiss1 component in pubertal KiDKO females. Since AVPV Kiss1 neurons have been shown to play an important role in the control of female puberty, as partial knockdown (<40%) of AVPV kisspeptin expression delayed pubertal maturation[43], and are essential for ovulatory induction[44], our data suggest the integral conservation of the Kiss1 neuronal system, including also the ARC, is relevant for the completion of puberty and attainment of fertility. In this context, partial knockdown of ARC kisspeptin has been shown recently to diminish the amplitude of the preovulatory LH surge, responsible for ovulation[43]; findings that are well aligned with our current data on the lack of LH surge induction in KiDKO female mice after sex steroid priming. Hence, the suppression of Kiss1/Kisspeptin in the ARC of pubertal KiDKO female mice may contribute also to the lack of first ovulation and persistent infertility in these animals.

In sum, we present herein evidence for an essential role of miRNA biogenesis in Kiss1 neurons for the completion of female puberty and maintenance of adult fertility in both sexes. In contrast, early stages of sexual maturation, the onset of puberty, and survival of Kiss1 neurons were fully preserved in mice with congenital ablation of Dicer in Kiss1 cells, denoting the dispensable (or compensable) role of miRNA-regulatory pathways and/or Kiss1 function in these phenomena. Intriguingly, congenital ablation of miRNA biosynthesis in Kiss1 neurons evoked consistent upregulation of key Kiss1 repressors, therefore supporting the importance of miRNA-mediated inhibition of repressive signals as a major mechanism for the precise control of Kiss1 expression and, thereby, reproductive function. Our data open up the possibility that alterations in miRNA biogenesis in Kiss1 neurons might be causative of late-onset forms of central hypogonadism, either spontaneous or associated with other co-morbidities, such as obesity or diabetes[45]. In addition, the differential impact of Dicer ablation between sexes and Kiss1 neuronal subpopulations highlights genuine differences in the roles of Kiss1 miRNA-machinery in the precise control of male and female reproduction.

## Methods
### Animals
Genetically modified C57Bl6 mice, engineered for conditional ablation of Dicer in Kiss1 and GnRH cells (KiDKO and GoDKO mice, respectively; see below) were bred in the vivarium of the University of Cordoba. The animals were housed under constant conditions of light (12-h light/dark cycles) and temperature (22 ± 2 °C). The day the litters were born was considered day 1 of age (PND1). Animals were weaned at PND-21 and provided with free access to tap water and fed ad libitum with a standard soy-free diet. The experiments and animal protocols included in this study were approved by the Ethical Committee of the University of Cordoba; all experiments were conducted in accordance with European Union (EU) normative for the use and care of experimental animals (EU Directive 2010/63/UE, September 2010).

### Generation of KiDKO and GoDKO mouse lines
Conditional elimination of Dicer enzyme in Kiss1 cells was accomplished by crossing mice driving Cre recombinase protein expression under the endogenous promoter of Kiss1 gene[46] with mice containing loxP sites flanking the exon 23 of the Dicer gene[47]; the resulting mouse line was named KiDKO, for Kiss1-specific Dicer KO, and lacks functional Dicer enzyme in Kiss1-expressing cells, therefore causing the absence of Dicer-dependent mature miRNAs specifically in these cells. A similar strategy was implemented for ablation of Dicer in GnRH cells: the Dicer-loxP mouse line was crossed with mice expressing Cre recombinase protein under the endogenous promoter of GnRH gene[48], thereby producing the GoDKO mouse line (for Gonadotropin-releasing hormone-specific Dicer KO).

In detail, heterozygous Kiss1[Cre/-] and positive GnRH-Cre[pos] mice were initially mated with Dicer[loxP/loxP] animals. The resulting genotypes, Kiss1[Cre/-]::Dicer[loxP/-] or GnRH-Cre[pos]::Dicer[loxP/-], were self-crossed to generate all the possible genotypic combinations. For the experiments, only Kiss1[Cre/-]::Dicer[loxP/loxP] (referred as KiDKO mouse) and GnRH-Cre[pos]::Dicer[loxP/loxP] (termed as GoDKO mouse) were used. In addition, Kiss1[-/-]::Dicer[loxP/loxP] and GnRH-Cre[neg]::Dicer[loxP/loxP] mice were used as controls of each line. Of note, for some immunohistochemical experiments in which CreGFP expression was required for the detection of Kiss1 neurons, Kiss1[Cre/-]::Dicer[loxP/-] mice were used as controls.

In addition, for the experiments of tracking Kiss1 neuronal survival and for isolation of Kiss1 neurons by FACS, a triple transgenic mouse line, expressing the Cre-dependent reporter (ROSA26)-YFP (Strain #006148; The Jackson Laboratory; Bar Harbor, ME) in Kiss1 neurons, was generated on the KiDKO background.

### Genotyping of KiDKO and GoDKO mouse lines
PCR analysis on isolated genomic DNA from ear biopsies was used to screen genotypes. Primers used to detect the presence or absence of Dicer-loxP allele were: DicerF1 5′-CCTGACAGTGACGGTCCAAAG-3′ and DicerR1 5′-CATGACTCTTCAACTCAAACT-3′. The loxP allele produces a 420-bp PCR product, whereas the wild-type allele results in a 351-bp band. An additional pair of primers, DicerF2 5′-ACTGAGGTAACTGAAAACCC-3′ and DicerDel2 5′-ATGTAGGTTAAAGCTGTTTG-3′ was used in order to detect the null allele, resulting from the recombination of loxP sites and, consequently, Dicer allele loss of function. Using this pair of primers, the recombinant (deleted) allele produces a 700-bp PCR product, whereas the wild-type allele results in an 820-bp band, whose identities were confirmed by direct sequencing (SCAI-Genomic Unit; University of Córdoba).

To detect the presence or absence of Cre recombinase under the Kiss1 promoter, a combination of three primers was used: 1614 5′-GACCTAGGCTCTGGTGAAG-3′, 1326 5′-GGCAAATTTTGGTGTACGGTCAG-3′, and WTRP 5′-GAGCCTCCAGTGCTCACAGC-3′. This primer combination amplifies a band of 337bp for the wild-type allele and 250 bp for the Cre gene. Primers for detection of Cre expression under the GnRH promoter were: CreF1 5′-CCTGGAAAATGCTTCTGTCCG-3′ and CreR1 5′-CAGGGTGTTATAAGCAATCCC-3′. A 400 bp PCR product denotes the presence of the Cre allele.

Amplification was carried out using the following protocol: denaturing for 5 min at 95 °C followed by 35 cycles, consisting of denaturing at 95 °C for 30 s, annealing (at 53 °C for Dicer-loxP, 60 °C for Kiss1-Cre, and 55 °C for GnRH-Cre) during 30 s and extension at 72 °C for 1 mine, using an iCycler iQ®5 (Bio-Rad Laboratories, Hercules, CA, USA). Samples were subsequently analyzed on 2% agarose gels.

### Validation of the KiDKO mouse line
While the GoDKO line has been previously characterized by our group[25], the KiDKO mouse line is newly-generated and had not been reported. To validate the specificity of conditional ablation of Dicer in Kiss1 cells in this line, three different approaches were used. First, PCR analyses at the hypothalamic regions containing Kiss1 neurons

(namely, ARC and POA, the later including AVPV, or whole hypothalamus) and cortex were done to validate the presence of the recombined Dicer allele, therefore denoting ablation of the exon 23 of *Dicer*. The hypothalamus was dissected immediately upon euthanasia of the animals, and dissection of the indicated hypothalamic regions was conducted, following previous protocols[49]. In addition, samples of the cerebral cortex were obtained from the same animals and used as a negative control due to the absence of Kiss1 neurons in this area. Samples from KiDKO and control mice were included in PCR analyses (Fig. S1a).

Although the sensitivity of PCR assays described above is very high, this procedure does not provide cellular resolution to document the ablation of Dicer, specifically in Kiss1 neurons. To solve this, two additional techniques were implemented. First, double in situ hybridization for *Kiss1* and *Dicer* exon 23 was carried out in the ARC region of young-adult (PND54), ovariectomized (OVX) KiDKO female mice and their respective OVX controls. To this end, BaseScope™ Duplex Assay was used in 20 μm frozen brain slices from KiDKO and control mice following manufacturer guidelines (BaseScope™ Detection Reagent Kit; ACDBio Cat No. 323800). Probes used for detection of *Kiss1* and *Dicer* exon 23 were BA-Mm-Kiss1-4zz-st (ACDBio custom made No. 715701) and BA-Mm-Dicer1-4z-C2 (ACDBio custom made No. 718101-C2). Brain slices were counter-stained with hematoxylin and 0,02% ammonium hydroxide, air-dried, and cover-slipped after the addition of Vectamount mounting medium (Vector Laboratories). Co-expression of both mRNA species was assessed by visualization in a Leica DM2500 microscope (Fig. S1b).

In addition, functional ablation of Dicer activity was evaluated by qPCR detection of Dicer mRNA (Fig. S1e) and the mature miRNA, let-7b-5p (Fig. S1c), in isolated Kiss1 neurons from KiDKO mice obtained by FACS (see below). In our validation analyses, let-7b-5p was chosen based on previous studies showing changes in hypothalamic expression of the let-7b/Lin28 system during pubertal maturation[50] and the abundant expression of members of the let-7 family of miRNAs in hypothalamic nuclei, including the ARC[51]. To this end, AVPV and ARC Kiss1 neurons populations were isolated separately from control and KiDKO mice by FACS, following an improved version of our previously validated protocol[26], as described below in the section *Isolation of Kiss1 neurons by fluorescent-activated cell sorting* (FACS). Verification of ARC Kiss1 neurons was conducted by qPCR analysis of the expression of *Kiss1* and *Tac2* mRNAs. In addition, contamination with other ARC neuronal populations, such as NPY/AgRP neurons, was discarded by confirmation of the absence of *Npy* expression in the isolated Kiss1 cells (Fig. S1c). Furthermore, to quantify the extent of Dicer ablation with single-cell resolution, ARC Kiss1 neurons from control and KiDKO mice were isolated individually by FACS and single-cell qPCR analyses of *Kiss1*, *Dicer*, and *Gapdh* were performed; the latter serving as a control to confirm cell collection in each well. In addition, the percentage of Dicer-expressing Kiss1 cells were calculated (Fig. S1d).

For mRNA expression analysis from isolated cells, RNA was reverse transcribed using the High Capacity cDNA RT Kit (4368814, Applied Biosystems), while for miRNA expression study, TaqMan MicroRNA Reverse Transcription Kit (4366596; Applied Biosystems) was applied, including the RT primer for let-7b (Assay ID: 000378), following the manufacturer instructions. Then, 2 μl of each RT product were amplified using the TaqMan Universal Master Mix II, according to the manufacturer's protocol (Thermo Fisher Scientific, Waltham, MA). qPCR was carried out on CFX96 Touch Real-Time PCR Detection System (Bio-Rad) using the following TaqMan Gene Expression Assays (Thermo Fisher): *Kiss1* (Assay ID: Mm03058560_m1), *Tac2* (Assay ID: Mm01160362_m1), *Dicer* (Assay ID: Mm00521731_m1), *Npy* (Assay ID: Mm01410146_m1), *let-7b* (Assay ID: 000378), and *Gapdh* (Assay ID: Mm99999915_g1); the later served as a housekeeping gene. The PCR conditions used were 1 cycle of polymerase activation at 95 °C for

10 min, followed by 40 cycles for group-isolated Kiss1 neurons, or 50 cycles, for single-cell expression analysis, denaturation at 95 °C for 15 s and annealing at 60 °C for 1 min. Relative expression of each gene respecting the housekeeping was calculated using the $2^{-\Delta\Delta Ct}$ method.

### Physiological measurements
**Pubertal maturation.** For phenotypic evaluation of pubertal maturation, somatic and reproductive indexes of pubertal development were monitored daily after weaning (at postnatal day 21; PND-21). These included body weight monitoring, and assessment of balano-preputial separation (BPS) in males, and vaginal opening (VO), in females, both considered external signs of puberty onset.

**Fertility index.** To evaluate fertility in control and KiDKO animals, male and female mice were crossed with wild-type mice of the opposite sex of proven fertility, and the percentage of animals being able to generate offspring was calculated.

**Estrous cyclicity.** Estrous cycles were monitored by daily vaginal cytology in adult female mice during 3–4 weeks. To this end, cells were collected by vaginal smears and deposited on a glass slide. After drying, cells were stained with toluidine blue and visualized in a light microscope for evaluation of the cycle stage, according to the presence of cell types characteristic of each phase of the estrous cycle[52].

### Evaluation of gonadal maturation
To evaluate the status of the reproductive organs, different histological parameters of male and female gonads were assessed in control and conditional null animals.

**Tissue processing.** After tissue collection, testes and ovaries, together with the oviduct and the tip of the uterine horn, were fixed for at least 24 h in Bouin solution and processed for paraffin embedding. The gonads were serially sectioned (10 μm-thick), stained with hematoxylin and eosin, and viewed under an Elipse 400 microscope for evaluation.

**Ovarian parameters.** Differences between control and conditional null mice in the maturation of the ovaries were assessed by morphometric analyses of the number of follicles at the different stages of development and the presence of corpora lutea; the latter being taken as an index of ovulation. Follicles were classified, according to the degree of development in resting follicles (RF), primary follicles (P1), secondary follicles (Sc), antral follicles, and atretic follicles, following published protocols for dating of ovarian maturation[53].

**Testicular parameters.** Testicular maturation was assessed according to the number of germs cells at the different stages of the spermatogenic cycle and the diameter of the seminiferous tubules; the latter being an indicator of the progression of spermatogenesis. The types of germ cells considered at the different stages of the spermatogenic process (from less to more differentiated) were: primary (pachytene) spermatocytes, round spermatids, considered as spermatids up to step 9 of the spermatogenic process, and elongated spermatids, from the step-10 onward. In addition, the proportion of tubules with dying germ cells was quantified as an index of atrophic seminiferous tubules with spermatogenic arrest. This classification was made attending to previous reference[54].

**Ovulatory capability.** To evaluate the ovulatory response in control and conditional null animals, a standard gonadotropic priming protocol was carried out, following a modified protocol from Jackson Lab. for superovulation. To avoid the possible confounding factor of normal ovulatory events in cycling females, immature animals were used. In brief, control and KiDKO female mice were injected at 1:00-4:00 pm of PND22 with PMSG (5 U/100 μl saline; Sigma-Aldrich Co.)

followed by a second injection with hMG-Lepori (5 U/100 μl saline; Angelini Pharma España, S.L.), 2 days after the first injection, at 11:00–12:00 a.m. Animals were euthanized 24 h later by an overdose of ketamine-xylazine and ovaries were collected and fixed in Bouin for histological processing. The number of oocytes released to the oviduct was quantified.

#### Preovulatory surge induction protocol

To evaluate the capability of KiDKO females to response to the positive feedback of gonadal steroids with a preovulatory-like LH surge, attributed mainly to AVPV Kiss1 neurons[1,27], a previously validated sex steroid-induction protocol was applied[55]. In brief, young females of 6–8 weeks old were bilaterally ovariectomized (OVX) and implanted with silastic capsules containing 5 μg/mL of estradiol, dissolved in sesame oil, to reproduce the estradiol levels exerting a central negative feedback. Then, daily vaginal smears were obtained to monitor the estrus cycle and ~9 days after, when all females were in diestrus, a subcutaneous injection of estradiol benzoate were given between 10–11 a.m., at a dose of 1 μg per 20 g BW to reproduce the positive feedback levels of estradiol. This protocol induces a preovulatory surge in the afternoon of the following day. For monitoring of LH changes, blood samples were obtained from tail-tip bleeding in the morning (10:00 and 10:30 a.m.) and during the afternoon—before and after the lights turn off (at 19:00 p.m.)—from 18:00 to 21:00 p.m. in intervals of 30 min. LH levels were measured by ELISA, as described below in the section Hormone Measurements.

#### Pharmacological and gonadectomy tests

To evaluate central responsiveness to different activators of the reproductive axis, pharmacological and gonadectomy tests were carried out in adult control and KiDKO animals. Pharmacological tests were done specifically in male mice in order to avoid the confounding factor of variations in gonadal hormones occurring in cycling females along the cycle.

Gonadotropin responses were studied after intracerebroventricular (icv) injection of Kisspeptin-10 (1 nmol/5 μl; Phoenix Pharmaceuticals, Inc) and NMDA (1 nmol/5 μl; Merck KGaA, Germany). In addition, to test the pituitary responsiveness to GnRH, GnRH (0.25 μg/100 μl; Merck KGaA, Germany) was injected intraperitoneally. For evaluation of the changes in gonadotropin release, blood samples were collected by jugular venipuncture, before cannulation of the animals, for evaluation of basal levels and after injection of the different drugs (15 min for Kp-10 and NMDA, and 30 min for GnRH). The different tests were carried out in the same mice, after washout periods of 5–7 days between tests for recovery of the animals. Doses and routes of administration were selected based on previous references[56,57]. Central (icv) administration of the compounds was done following standard procedures of cannulation[58]. In brief, cannulas (Intra-medic polyethylene tubing; Becton Dickinson) were inserted to a depth of 2 mm beneath the surface of the skull, with an insert point at 1 mm posterior and 1.2 mm lateral to Bregma, according to a mouse brain atlas. Drugs were dissolved in 0.9% saline.

For evaluation of gonadotropin response to gonadectomy, blood samples were collected by jugular venipuncture in male mice before and 2 weeks after bilateral orchidectomy, as described elsewhere[39].

#### Hormone measurements

For hormone assays, three different protocols were carried out. For evaluation of gonadotropin levels in single-point determinations, serum samples were measured by Radioimmunoassay (RIA), while continuous evaluation of LH pulsatility was analyzed by a high-sensitive ELISA. Finally, steroids levels were assessed in serum samples from single-point determinations by liquid chromatography–tandem mass spectrometry (LC-MS/MS).

**RIA assay.** Blood samples for LH and FSH measurements were obtained by cardiac puncture in infantile and pubertal animals after overdose of ketamine-xylazine or jugular venipuncture in adults. After blood collection, serum was separated by centrifugation at 1620 g for 30 min and maintained at −20 °C until used for hormone assays.

Serum LH and FSH levels were determined in a volume of 25–50 μl using a double-antibody method and RIA kits supplied by the National Institutes of Health (A. F. Parlow, National Institute of Diabetes and Digestive and Kidney Diseases National Hormone and Peptide Program -NIDDK-NHPP; Torrance, CA) following previously validated protocols[59]. In brief, Rat LH-I-10 and FSH-I-9 were labeled with 125I using Iodo-gen tubes, following the instructions of the manufacturer (Pierce, Rockford, IL). Hormone concentrations were expressed using reference preparations, LH-RP-3 and FSH-RP-2, as standards. Intra- and inter-assay coefficients of variation were, respectively, <8 and 10% for LH and <6 and 9% for FSH. The sensitivity of the assay was 5 pg/tube for LH and 20 pg/tube for FSH. The accuracy of hormone determinations was confirmed by the assessment of mouse serum samples of known hormone concentrations used as external controls.

**Pulsatile LH measurement.** For evaluation of LH pulsatility, mice were habituated to daily handling for a period of minimum 3 weeks. Four microliters of blood were collected by tail-tip bleeding every 5 min for 3 h. The blood was directly diluted in 46 μl of PBS-Tween0.05%, frozen on dry ice, and stored at −80 °C. The LH content of the blood sample was measured by a sensitive LH sandwich ELISA[60]. A 96-well high-affinity binding microplate (Corning) was coated with 50 μl of the monoclonal antibody, anti-bovine LHβ subunit, 518B7 (L. Sibley; University of California, UC Davis) at a dilution of 1:1000 (in PBS: $Na_2HPO_4/NaH_2PO_4/NaCl$ 0.1 M, pH 7.4) and incubated overnight at 4 °C. The next day, the plate was firstly incubated with 200 μl of a blocking buffer (milk powder (5%) in PBS-Tween0.05%) for 2 h at room temperature (RT), then 50 μl of the blood samples and the LH standard curve were incubated for 2 h at RT. Standards were made from rLH-RP3 (supplied by Dr. A.F. Parlow) and used at twofold serial dilutions starting at 1 ng/ml to 0.0078125 ng/ml in PBS-Tween. Fifty microliters of the detection antibody (rabbit LH antiserum, AFP240580Rb; provided by NIDDK-NHPP) diluted at 1:10000 (in the blocking buffer) were incubated for 1.5 h at RT, followed by incubation of 50 μl of horseradish peroxidase-conjugated antibody (goat anti-rabbit; Vector Laboratories) at a dilution of 1:1000 in 50% blocking buffer and 50% PBS during 1.5 h at RT. Finally 100 μl of o-Phenylenediamine, diluted in citrate buffer (pH 6) with 0,1% $H_2O_2$, were added for 30 min at RT. The reaction was stopped with 50 μl of HCl (3 M), and the plate was read at a wavelength of 490 nm (and at 650 to detect the background). The LH concentrations were measured by interpolating the OD values of unknown samples against a nonlinear regression of the LH standard curve. The assay sensitivity was 0.002 ng/ml.

An LH pulse was identified for each point, presenting an increased value of a minimum of 125% from the baseline nadir to peak. The basal level was determined as the mean of the 5 lowest values recorded in 3 h. This criterion is based on recent evidence demonstrating a correlation between synchronized episodes of calcium activity in ARC Kiss1 neurons with pulsatile LH secretion[61].

**Sex steroid measurements.** Blood samples for sex steroid measurements were obtained by cardiac puncture in peripubertal (4-week-old) and adult (>2-month-old) mice of control, KiDKO, and GoDKO phenotypes, after overdose of ketamine-xylazine; serum samples were separated by centrifugation. For pubertal animals, due to the reduced volume of serum obtained, samples were pooled at a volume of 200 μl to improve the detection sensibility in this age-window of reduced steroid levels. In contrast, in adult mice, individual measures were

taken in 150–200 µl of serum samples. Protocols of LC-MS/MS for accurate measurement of testosterone (in males), estradiol (in females) and progesterone (in both sexes) were applied to serum samples, as described in detail recently[62].

## Brain histological analyses

For the detection of the number of Kiss1 neurons, in situ *Kiss1* expression and kisspeptins and NKB contents, different histological protocols were carried out.

**Tissue processing.** For immunohistochemical analyses, mice were euthanized with an overdose of ketamine-xylazine and perfused intracardially with saline (0.9% NaCl) followed by 4% PFA in PBS (pH 7.4). Fixed brains were collected and immersed in 30% sucrose and 0.01% sodium azide in PBS at 4 °C for 2–4 days. Next, three sets of coronal (30-mm-thick) sections were cut in a freezing microtome (Leica CM1850 UV) and stored at −20 °C in cryo-protectant. For immunodetection of the different proteins, one set of sections encompassing the whole hypothalamus was used for each animal, and standard procedures for single-label immunohistochemistry were performed (see below).

For in situ hybridization (ISH) analyses, fresh brains were collected and frozen in dry ice following previous protocols[63]. Five sets of coronal (20-mm-thick) sections were generated and mounted on Super-Frost Plus slides (Thermo Fisher Scientific). Standard procedures of tissue collection were applied, starting on a fixed coordinate in the rostral hypothalamic area, to encompass equivalent areas of the anterior (including the POA and AVPV) and medio-basal (including the ARC) hypothalamus, where GnRH and Kiss1 neurons are abundantly located. The samples were stored at −80 °C until ISH analyses.

**Immunohistochemistry.** For protein detection in brain slices, one set of the free-floating section was washed in Tris-buffered saline (TBS; pH 7.6) at RT with gentle agitation to eliminate the cryo-protectant. Next, endogenous peroxidase was blocked by incubation with $H_2O_2$-methanol-TBS, and the sections were washed in TBS and incubated for 72 h at 4 °C with the primary antibody for kisspeptin (primary rabbit anti-rat/mouse kisspeptin antibody 1:20000; AC#566; a gift from Dr. Alain Caraty, PRC-INRA, 37380 Nouzilly, France), NKB (primary rabbit anti-pro-NKB 1:5000; IS39; a gift from Dr. Philippe Ciofi, INSERM U 1215, 33077 Bordeaux, France) or CreGFP and YFP (primary chicken anti-GFP antibody 1:40000; Abcam; Cat. Ab13970) detection. After this step, brain sections were washed again in TBS and incubated 90 min at RT with the corresponding secondary antibody (Biotinylated Donkey anti-rabbit 1:500; Code 711-066-152 or Biotinylated Donkey anti-chicken, 1:500; Code 703-066-155; Jackson ImmunoResearch Europe Ltd., UK). Next, brain slices were washed in TBS and incubated for 90 min at RT with Vectastain Elite ABC-HRP Kit (Vector Laboratories). Finally, brain sections were washed and protein detection was revealed using glucose-oxidase plus nickel-enhanced diaminobenzidine hydrochloride method[64]. Brain slices were mounted in glass slides, air-dried and cover-slipped, after processing with ascending concentrations of alcohol, xylene, and the addition of Eukitt mounting medium (Merck KGaA, Germany).

The total number of YFP and CreGFP of AVPV Kisspeptin-positive cells were counted by visualization under a Leica DM2500 microscope. Kisspeptin and pro-NKB-immunoreactivity was quantified by densitometry using ImageJ (NIH). In brief, pictures containing the complete ARC were binarized. The area containing kisspeptin and pro-NKB immunoreactivity was quantified in pixels and converted to $mm^2$ using the reference of a Microscope Glass Stage Micrometer Calibration Slide.

**In situ hybridization.** Brain expression of *Kiss1* and *GnRH* mRNA was assessed by in situ hybridization using $P^{33}$-riboprobes following a well-validated protocol[65]. A specific antisense riboprobe for the detection of mouse *Kiss1* or *GnRH* mRNA was generated according to a validated protocol[65]. Primer sequences for riboprobes generation are detailed in Suppl. Table 1. A single set of sections was used for ISH (adjacent sections 100-µm apart). These tissue sections were fixed in 4% PFA, acetylated in triethanolamine buffer, dehydrated in increasing concentrations of ethanol, and delipidated with chloroform. After these steps, hybridization with *Kiss1* or *GnRH* riboprobes was performed for 16 h at 55 °C. Riboprobes were diluted in hybridization buffer to a final concentration of 0.03 pmol/ml along with yeast tRNA. After hybridization, slides were washed, treated with RNase-A, and dehydrated in increasing ethanol series[65]. Finally, slides were dipped in Kodak Auto-radiography Emulsion type NTB (Eastman Kodak; Rochester, NY) and exposed for 2–3 weeks at 4 °C in dark. After this period, the sections were developed and fixed following the manufacturer's instructions (Kodak; Rochester, NY). Then, slices were cover-slipped with Sub-X mounting medium (Leica). For analysis, 30–36 sections from each animal (5–6 slides; 6 sections/slide) were evaluated. Slides were read under dark-field illumination with custom-designed software, enabled to count the total number of cells (grain clusters) as well as the number of silver grains/cell. Cells were counted as positive when the number of silver grains in a cluster exceeded that of the background.

## Isolation of Kiss1 neurons by fluorescent-activated cell sorting

To isolate Kiss1 neurons, a triple transgenic mouse line expressing (ROSA26-YFP) Cre-dependently in the KiDKO background was used. For experiments, Kiss1[Cre/−]::Dicer[loxP/loxP]::YFP[loxP/−], namely KiDKO-YFP mice, and Kiss1[Cre/−]::Dicer[loxP/−]::YFP[loxP/−], namely control-YFP mice, were used. Kiss1 neurons populations from the AVPV and ARC were isolated separately from hypothalamic tissue blocks containing the POA and the medio-basal hypothalamus (MBH), respectively. After micro-dissection, POA and MBH tissue blocks were enzymatically dissociated using a Papain Dissociation System (Worthington, Lakewood, NJ) to obtain single-cell suspensions. FACS was performed on these suspensions with a FACS Aria III Sorter (BD Biosciences), using FACSDiva 8.0 software (BD Biosciences). Data were analyzed using the Kaluza 2.0 software (Beckman Coulter). First, cellular debris was excluded by gating cells for forward scatter (FSC) area and side scatter (SSC) area, and later, aggregated cells were excluded by selecting singlets by plotting FSC area vs FSC height. Finally, the sort decision was based on measurements of YFP fluorescence (excitation: 488 nm; detection: YFP bandpass 530/30 nm, autofluorescence bandpass 780/60 nm) by comparison with cell suspensions from cortex, which does not harbor YFP-positive neurons. A total of 229 ± 37 AVPV and 574 ± 43 ARC YFP-positive neurons were collected per animal on 10 µL of lysis buffer (0.1% Triton X-100 and 0.4 unit/µl RNAsin, Promega). For the individual isolation of ARC Kiss1 neurons, 17 ± 1 individual neurons per animal were individually sorted on 96-well plates containing 10 µL of lysis buffer per well.

## Quantitative qPCR analyses of Kiss1 transcriptional repressors and activators

To evaluate gene expression levels of previously reported Kiss1 promoter repressors and activators, qPCR assays were performed in FACS-isolated AVPV and ARC Kiss1 neurons from peripubertal control and KiDKO mice (PND28). Gene Expression Assays (Applied Biosystems) used were: *Mkrn3* (Mm00844003_s1), *Sirt1* (Mm01168521_m1), *Eed* (Mm00469660_m1), *Cbx7* (Mm00520006_m1), *Yy1* (Mm0045 6392_m1), *Eap1* (Mm07300240_s1), *Cux1* (Mm01195598_m1), *Ttf1* (Mm00657018_m1), and *Mll1* (Mm01179235_m1). *Gapdh* (Mm999 99915_g1) was used as housekeeping gene. First, RNA from isolated Kiss1 neurons was DNAse-treated following the manufacturer's protocol (M6101; RQ1 RNase-Free DNase; Promega Corporation, Madison, WI) and reverse transcribed using the High Capacity cDNA RT Kit (4368814, Applied Biosystems). Prior to qPCR, cDNA was linearly

pre-amplified using the manufacturer's protocol of 14 cycles for the Taqman PreAmp Master Mix Kit (4391127, Applied Biosystems), and then, qPCRs were performed and analyzed as described in previous sections.

## Statistics and biostatistics analyses

Data were expressed, as the mean ± SEM. A normalization test was carried out in the groups of data and two-sided Student $t$-test for parametric or a two-sided Mann–Whitney for nonparametric was applied, consequently. For analysis of sex steroid levels, Kruskal–Wallis nonparametric test were carried out for comparisons between control, KiDKO and GoDKO mice. Results were analyzed using Prism GraphPad 9.0 software (GraphPad Software, Inc.). The significance level was set at $P \leq 0.05$ and asterisks indicate statistical significance. As a general rule, sample sizes were selected based on our previous experience with studies addressing neuroendocrine regulation of puberty and the reproductive axis, assisted by power analyses performed using values of standard deviation that we usually obtain when measuring parameters analogous to those examined in this study. Based on those calculations, minimal group sizes of $n = 6$ animals per group were established, unless otherwise indicated due to operational reasons, as analyses using these sample size should provide at least 80% power to detect effect sizes using the tests indicated above, with a significance level of 0.05. In general, for physiological experiments, group sizes largely exceeded this threshold. However, based on standard procedures, while phenotypic and hormonal analyses were applied to all available samples, more complex molecular/histological analyses in these experiments were conducted in a representative subset of randomly assigned samples from each group. Details on actual sample sizes for all determinations are provided in the corresponding figure legends. As general procedure, the investigators directly performing the experimentation involving physiological/molecular determinations were not blinded to the group allocation, but primary data analyses conducted by senior authors were conducted independently to avoid any potential bias.

Finally, bioinformatic analyses were applied to seek for putative seed regions of miRNA regulators in the 3′-UTR of *Mkrn3*, *Cbx7* and *Eap1* genes, since these transcripts were upregulated in Kiss1 cells from our KiDKO mice. To this end, the tools provided by TargetScan (http://www.targetscan.org/) and TarBase v8 (http://www.microrna.gr/tarbase) were used. As most salient findings, the 3′-UTR of *Mkrn3* and *Cbx7* were found to possess predicted seed regions for members of the miR-30 family.

## Reporting summary

Further information on research design is available in the Nature Research Reporting Summary linked to this article.

# Data availability

The authors confirm that the data supporting the conclusions of this study are included in this published article and its supplementary files (including source data files). Any additional data are available from the corresponding authors, upon reasonable request. Source data are provided with this paper.

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

## Acknowledgements

This work was supported by grants BFU2014-57581-P, BFU2017-83934-P, and PID2020-118660GB-I00 (to M.T.-S.—Ministerio de Economía y Competitividad, Spain; co-funded with EU funds from FEDER Program); Project PIE-00005 (to M.T.-S) and grants PI16/01243 and PI19/00257 (to J.R.—Instituto de Salud Carlos III, Ministerio de Sanidad, Spain; co-funded with EU funds from FEDER Program); Project P12-FQM-01943 (to M.T.-S.—Junta de Andalucía, Spain); Project REP-655232 (ReprObesity; to M.T.-S.–European Union); and COST Action BM1105 (to M.T.-S. and V.P.–European Union). CIBER Fisiopatología de la Obesidad y Nutrición is an initiative of Instituto de Salud Carlos III, Spain. The authors are indebted to Dr. Esther Peralbo, Head of the Cytometry Unit of IMIBIC, for her superb assistance in the optimization and implementation of protocols of FACS for Kiss1 neuronal isolation. The assistance of Profs. Catherine Dulac and Robert A. Steiner for providing the parental Kiss1$^{Cre/-}$ and GnRH-Cre$^{pos}$ mouse lines is cordially appreciated.

## Author contributions

J.R. was co-responsible for study design, conducted and coordinated the physiological experiments and analytical procedures, evaluated and discussed the data, was responsible for figure preparation, and initial drafting of the manuscript; M.R.-C. assisted in data generation and analysis, and was actively involved in data representation and analysis; F.R.-P. actively participated in the conduction of the various experimental studies, including prominently ISH and histochemical studies, as well analytical procedures, with the support and assistance of R.O., M.J.S.-P., J.M.R.-R., A.B., V.H., I.V., and C.P.-L., who all discussed the data; V.S. actively contributed to the optimization of Kiss1-cell isolation using FACS, while M.J.V. and M.S.A. actively participated in the generation, discussion, and interpretation of experimental results; C.O., V.P., and M.P. provided essential reagents (V.P. and M.P.) and sex steroid hormone measurements (C.O.), participated in study design and actively discussed the data; L.P. and F.G. participated in study design and analytical procedures, including hormonal analyses (LP) and histological studies (FG), and actively discussed the data; MTS designed and co-supervised the study, and contributed also to analyze and discuss the data. J.R. and M.T.-S. wrote the manuscript, which was initially reviewed by all senior authors, and later by the rest of authors. All the authors take full responsibility for the work. J.R. and M.T.-S. are both senior and corresponding authors of this study.

## Competing interests

The authors declare no competing interests.

## Additional information

**Supplementary information** The online version contains

supplementary material available at https://doi.org/10.1038/s41467-022-32347-4.

