## [Peer Review File · Nature Communications]

Reviewers' Comments:

Reviewer #1:

Remarks to the Author:

In this very interesting study, Roa et al perform targeted deletion of Dicer in Kiss1-expressing neurons (cells) and identify acquired hypogonadism in both sexes, with normal puberty onset and fertility in males before subsequent development of hypogonadotropic hypogonadism, and normal puberty initiation (vaginal opening) in females but failure to complete puberty, absence of estrous cyclicity, infertility and hypogonadotropic hypogonadism. They further show that these changes reflect decreases in ARC Kiss1 (GFP) expression by 2 weeks and Kisspeptin and NKB immunoreactivity by 4 weeks in both males and females, and with changes in AVPV Kiss1 immunoreactivity following somewhat later. Studies of Kiss1 neuronal survival show reductions in Kiss1 neurons by 4 weeks of age in males (the earliest time point assessed) but not until 2 months in the female ARC and until 4 months in the female AVPV Kiss1 neurons. Overall, these studies point to the importance of miRNAs in Kiss1 neurons in the regulation of the HPG axis, with earlier impact on female puberty and reproduction despite the later reduction in Kiss1 neurons in females compared to males. Moreover, the effects are manifest prior to substantial loss of Kiss1 neurons.

1. Do the females have first estrus?
2. Are the authors certain that Dicer deletion occurs congenitally? Have they documented at what point during pre- (or post-) natal Dicer ablation occurs?
3. Related to comment 2 above, at what age were the studies in Figure S1 (to validate the KiDKO mouse line) done? Why were there no differences in Kiss1 and Tac2 expression in Figure S1C? Certainly, by comparison, Kiss1 mRNA seems to be reduced in KiDKO mice in Figure S1B.
4. What are the mechanisms of the effects of Dicer deletion in Kiss1 neurons? Have the authors had an opportunity to assess miRNAs present in Kiss1 neurons in these mice?
5. It is worth noting that some of the comparisons between KiDKO mice and controls are made in the context of differences in sex hormone milieu. For example, In Figure 1, the LH pulses are measured in eugonadal control mice but in hypogonadal KiDKO mice. It might be more relevant to perform these LH pulsatility analyses in GDX male and female KiDKO and controls so that the sex steroid milieu is comparable and the LH pulsatility more closely reflects intrinsic hypothalamic activation of GnRH neurons.

Minor comments:

6. The panels in Figure S4 could be re-ordered to align with the order in which they are discussed in the text (GnRH, GDX, KP, NMDA).
7. There is no panel E in Figure S5 (it skips from D to F).

This is a very interesting and elegant manuscript in which targeted deletion of Dicer from Kiss1 neurons results in acquired loss of Kiss1 neurons and acquired hypogonadotropic hypogonadism, pointing to the importance of miRNAs in Kiss1 neurons and suggesting a role in acquired forms of central hypogonadism. The varying impacts on males and females, as well as on ARC vs AVPV Kiss1 neurons, further highlight sex differences in regulation of puberty and reproduction as well as distinctions between ARC and AVPV Kiss1 neuronal populations.

This work is impactful. It is an editorial decision as to whether this is sufficient for publication in Nature Communications, or whether additional data (e.g., the mechanisms by which Dicer deletion results in hypogonadotropic hypogonadism, such as the miRNAs implicated) is warranted. Another question that is not fully answered is whether the acquired onset of hypogonadotropic hypogonadism is the result of a delay in the deletion of Dicer in this model, or whether Dicer is indeed deleted congenitally as proposed, but there is a delay in the impact on Kiss1 neuronal survival.

Reviewer #2:

Remarks to the Author:

The manuscript describes the effect of ablation of Dicer in Kiss1 neurons in the hypothalamus on the puberty and fertility in mice. The kisspeptin neuron number, the expression level of kiss1 gene, and the survival of kisspeptin neurons are all ruined by the conditional knockout of Dicer.

The manuscript is interesting and well written, yet I only have some minor issues needed to be addressed detailed below.

1, Line161-163, "adult KiDKO mice displayed a preserved number of GnRH-expressing neurons in POA (Fig. S3A, C)", Why the inhibited kiss1 gene expression did not affect the expression of GnRH? Were there any other ways by which kisspeptin influenced puberty or fertility bypassing GnRH neurons? Could the author provide some other possible mechanisms mediating kisspeptin and puberty and fertility?

2, Fig S1A, what was the PCR template, what was the age of the DNA providers? I think it's essential to clarify the time point of the Cre enzyme beginning to work, therefore, the age of mice when Dicer ablation in Kiss1 neurons came into being could be confirmed.

3, Line 210-214: The trends of increase of uterus and ovarian weights during 2w to 4w between the KiDKO and GoDKO female mice were different, what made the difference?

4, Why were the KiDKO female mice infertile though their ovulation seemed normal?

5, Line 241 "(Fig S6A,D, G)" could be "(Fig S6A)".

6, Before 4 weeks of age, the kisspeptin expression in the ARC was ruined by Dicer ablation more intensively than in the AVPV area, could you please explain such difference?

7, Dicer ablation in kisspeptin neurons suppressed the number of kisspeptin neurons and decreased the survival of these neurons after 4 weeks according to Fig3 and Fig 6. Could it be concluded that the survival of kisspeptin neurons depended on or independent of the kisspeptin expression?

8, Lin2 532-536: In "material and methods" section, pubertal maturation were evaluated by body weight monitoring, assessment of BPS (in male) and VO (in female). However, the results of BPS and VO of KiDKO mice were not provided, Were there any differences in these pubertal phenotypes between KiDKO and control mice?

Reviewer #3:

Remarks to the Author:

Nature Communications MS

NCOMMS-21-13176

Comments to Authors

In this MS the Authors have generated mice with a deletion of Dicer in Kiss1 expressing cells and analyzed the neuroendocrine and reproductive phenotypes during different stages of HPG axis development and maturation. For some of the presented data, they compared Dicer cKO in Kiss1 neurons with Dicer cKO in GnRH neurons. While the phenotypic analysis is extensive and well done, the MS suffers from lack of any mechanistic data that support their conclusions. Surprisingly, the Authors have not included the extent to which Dicer is deleted in the specific cell populations. Other comments are given below:

- The title is confusing. Could be reworded and also the Authors may want to consider to include Dicer deletion in GnRH cells also.
- It would be nice to show how much Dicer is deleted in the Cre+ cells by qPCR, similarly in FACS sorted cell populations, it would be important to show the levels of Kiss1, instead of qualitatively showing in situ expression data.
- The Authors have not mentioned anything about which miRNAs target Kiss1 mRNA and if these are suppressed or increased or unchanged in Dicer cKO mice in Kiss+ cells in both ARC and AVPV populations.
- Line 112: What do the Authors mean "fine control of the reproductive brain"?
- Line 117: it should be Mendelian ratio; not Mendelian rate.
- Line 136 & Line 138: shouldn't it be conditional null animals? And conditional null females? Not null animals and null females!

- At several places in the Results and Discussion, the Authors have made qualitative statements without actually mentioning how much suppression or increase was noted in expression of genes and whether these changes were statistically significant. These need to be fixed throughout the text.
- Line 179: could this be a direct pituitary effect?
- Are serum levels of steroids measured (estradiol, progesterone and testosterone) in cKO mice at different ages?
- The Authors argue repressors are activated in deleted cells and therefore Kiss1 protein was suppressed. But, there is no data provided to support this. It is essential for the Authors to check RNA/protein expression of repressors or activators and miRNAs that target these in Dicer deleted cells to come up with any mechanistic explanation for the phenotypes they observed.

Reviewer #1**Comments**

“In this very interesting study, Roa et al perform targeted deletion of Dicer in Kiss1-expressing neurons (cells) and identify acquired hypogonadism in both sexes, with normal puberty onset and fertility in males before subsequent development of hypogonadotropic hypogonadism, and normal puberty initiation (vaginal opening) in females but failure to complete puberty, absence of estrous cyclicity, infertility and hypo-gonadotropic hypogonadism. They further show that these changes reflect decreases in ARC Kiss1 (GFP) expression by 2 weeks and Kisspeptin and NKB immunoreactivity by 4 weeks in both males and females, and with changes in AVPV Kiss1 immunoreactivity following somewhat later. Studies of Kiss1 neuronal survival show reductions in Kiss1 neurons by 4 weeks of age in males (the earliest time point assessed) but not until 2 months in the female ARC and until 4 months in the female AVPV Kiss1 neurons. Overall, these studies point to the importance of miRNAs in Kiss1 neurons in the regulation of the HPG axis, with earlier impact on female puberty and reproduction despite the later reduction in Kiss1 neurons in females compared to males. Moreover, the effects are manifest prior to substantial loss of Kiss1 neurons.”

Response: We very much appreciate the detailed summary of our major findings and the overall positive comments of the referee. For specific review actions in response to his/her queries, see our comments below.

Specific comments

1. *“Do the females have first estrus?”*

Response: KiDKO female mice failed to display first estrus, neither they had other signs (e.g., histological) of first ovulation. This notion is clearly stated in the revised manuscript, see page 8, lines 223-224.

2. *“Are the authors certain that Dicer deletion occurs congenitally? Have they documented at what point during pre- (or post-) natal Dicer ablation occurs?”*

Response: The referee makes a relevant point. While in the original paper we had provided evidence for

effective Cre-mediated recombination in peripubertal KiDKO mice, in hypothalamic areas known to harbor Kiss1 neurons, we had not addressed when this deletion occurs. We have now directly analyzed this issue by conducting PCR assays for detection of Dicer recombination in the hypothalamus of KiDKO mice and their corresponding controls at an earlier age-point. Due to operational reasons, and given the fact that early stages of postnatal maturation seems to be preserved in our KiDKO mouse line, we analyzed the occurrence of recombination in neonatal mice, i.e., at postnatal day-1 (PND1). As shown in new Suppl. Fig. S1a, evidence for effective recombination of Dicer in the hypothalamus was obtained as early as PND1. This finding, together with the fact that Cre activity is driven by the endogenous Kiss1 promoter, which is reportedly active before birth in mice, with detectable *Kiss1* mRNA expression in the ARC as soon as embryonic day 13 (Knoll et al. *Front Endocrinol* 2013;4:140; PMID: 24130552), argue in favor of an early, congenital ablation of Dicer in Kiss1 neurons, in line with our initial expectations. As complementary comment, our qPCR analyses in individual Kiss1 neurons isolated from our Dicer^{-/-}:YFP mouse line, using FACS, conclusively demonstrate effective conditional ablation of Dicer from Kiss1 neurons in our model at the two postnatal ages (PND14 and PND28) tested. All these features reinforce the validity of our model to induce effective ablation of Dicer in Kiss1 neurons during early developmental stages. For further details, see the new Suppl. Fig. S1 and the revised manuscript (page 5, lines 120-121 and 131-135; and page 37, line 1121-1124).

3. “Related to comment 2 above, at what age were the studies in Figure S1 (to validate the KiDKO mouse line) done? Why were there no differences in *Kiss1* and *Tac2* expression in Figure S1C? Certainly, by comparison, *Kiss1* mRNA seems to be reduced in KiDKO mice in Figure S1B”

Response: The referee is right; there was some missing information about the experimental procedures related with some analyses in this figure, that could be misleading and confusing for the reader. In one hand, PCR recombination analyses in hypothalamic (and reference) tissues and qPCR studies in isolated Kiss1 neurons were performed peripubertal mice; these have been now completed with additional age-points for analysis of recombination events (PND1) and Kiss1 neuron qPCR analyses (on PND14 and 28; see our response to point#2 of this referee). On the other hand, ISH analyses using BaseScope® were conducted as kind of proof-of-principle, aiming to provide conclusive evidence for the effective ablation of Dicer from Kiss1 neurons. Since we were aware that BaseScope® may suffer from limited sensitivity, and assuming that Kiss1 expression declines in our KiDKO mouse line, we decided to use a model of elimination of negative feedback, i.e., ovariectomized (OVX) female mice, as a means to enhance endogenous Kiss1 expression and discard any potential “false negative” due to limited detection of Kiss1 neurons. For this reason, analyses were conducted in OVX mice at PND54. Accordingly, the comparison of former Suppl. Fig. S1B between control and KiDKO mice revealed a notable drop of Kiss1 mRNA expression in our conditional line, due to the fact that in control mice, OVX caused a rise in Kiss1 expression in the ARC. In contrast, in KiDKO mice, due to their age (>PND54) and lack of response to OVX, endogenous Kiss1 expression remained low. We are sorry if this aspect was not sufficiently clear in the original submission; this contention has been made clearer in the revised manuscript (see page 5, lines 125-126; page 20, lines 564-565; and page 37, line 1127). Additionally, to better support the validation of the KiDKO mouse line with single-cell resolution, we have isolated individual ARC Kiss1 neurons from KiDKO and control animals by FACS to perform qPCR analysis of Dicer and Kiss1. These data confirmed that the percentage of Dicer-expressing Kiss1 neurons was significantly reduced in the KiDKO mice (see Suppl. Fig. S1d and page 5, lines 131-133).

On the other hand, the information included in the original Suppl. Fig. S1C did not include replicates as to provide quantitative analyses, but rather intended to document that in isolated ARC Kiss1 neurons, denoted by detectable expression of Kiss1 and *Tac2*, the expression level of an abundant miRNA, namely *let-7b*, was markedly suppressed, in line with the conditional ablation of Dicer. While we believe that figure panel served this purpose, we acknowledge the limitation posed by the lack of quantitative measurements. Accordingly, and given the efforts devoted during the revision to include qPCR data from Kiss1 neurons isolated by FACS, we have incorporated in the revised Suppl. Fig. S1c quantitative analyses of the expression levels of Kiss1, *Tac2*, *Npy* (as negative control of specificity of isolated Kiss1 cells) and *let-7b*. These data confirm our original findings, showing a significant drop in *let-7b* levels in Kiss1 cells from our KiDKO mouse line at puberty (see new Suppl. Fig. 1c and page 5, lines 127-131).

In addition, our qPCR analyses in isolated Kiss1 neurons from gonadal intact, pubertal control and KiDKO mice, have unambiguously documented also, in a direct quantitative manner, the effective suppression of Dicer mRNA expression in Kiss1 neurons of our KiDKO mouse line, of both sexes (see Suppl. Fig. S1d,e). Thus, we believe that our dataset solidly documents the validity of our experimental model, which is now fully supported by genomic DNA, in situ RNA hybridization and Kiss1-cell qPCR analyses; for further details, see the revised Suppl. Fig. S1 and page 5, lines 115-135).

4. “What are the mechanisms of the effects of Dicer deletion in Kiss1 neurons? Have the authors had an opportunity to assess miRNAs present in Kiss1 neurons in these mice?”

Response: As pointed out also by the editor, this is a relevant aspect of our study, which was admittedly underdeveloped in our original submission. Accordingly, substantial revision efforts have been focused on providing molecular evidence for the putative mechanisms underlying the phenotype of our KiDKO mouse line. For these analyses, our line of reasoning was that, since Dicer ablation should remove all mature miRNAs in a cell-specific manner, high-throughput detection of the miRNAs present or absent in Kiss1 neurons from KiDKO mice might be less informative. Similarly, since miRNAs are predominantly repressors of mRNA stability and/or translation, and Kiss1 mRNA and protein were actually suppressed in KiDKO mice, we assumed that ablation of direct miRNA regulators of Kiss1 gene are less likely to contribute to this molecular phenotype, but rather elimination of miRNAs acting as suppressors of transcriptional repressors of Kiss1 gene would be likely to play a major role.

On this basis, we screened the literature and selected a panel of reported transcriptional regulators of Kiss1, including putative repressors or activators. For details of the major transcriptional repressors and activators of Kiss1 selected for our analyses, see our revised paper (page 11, lines 318-322). Of note, these qPCR analyses were conducted in isolated Kiss1 neurons obtained by FACS from our KiDKO mouse line, thus providing a notable level of precision to our studies.

Remarkably, these expression analyses have allowed us to delineate a tenable mechanism for the observed suppression of Kiss1 in our KiDKO mouse, as major determinant for its phenotype, since three important transcriptional repressors of Kiss1, namely Mkrn3, Cbx7 and Eap1, were found to be upregulated in male and female KiDKO mice. Of note, in a recent unrelated study, we have documented that Mkrn3 is negatively regulated by miR-30b (Heras et al., PLoS Biology 2019;17:e3000532), which also has a putative seed region in the 3'-UTR of Cbx7 (new bioinformatic analyses included in the revised paper; see page 13, lines 369-371 and 373-376; and page 29, lines 845-849). Therefore, our present study not only discloses the need of proper miRNA biosynthesis for adequate functioning of Kiss1 neurons, but demonstrates also the physiological relevance of novel miRNA-regulatory pathways of Mkrn3 and Cbx7, as well as possibly Eap1, in the control of Kiss1 expression, as principal component for the regulation of reproductive capacity.

We do believe that addition of this relevant piece of data endows our paper with a substantial mechanistic dimension, in line with the request of the referees and editor. Accordingly, these issues have been extensively discussed in the revised paper; see page 2, lines 41-42 & 45-46; page 11, lines 314-331; and page 12, lines 353-365, cont. page 13, lines 366-383.

5. “It is worth noting that some of the comparisons between KiDKO mice and controls are made in the context of differences in sex hormone milieu. For example, In Figure 1, the LH pulses are measured in eu-gonadal control mice but in hypogonadal KiDKO mice. It might be more relevant to perform these LH pulsatility analyses in GDX male and female KiDKO and controls so that the sex steroid milieu is comparable and the LH pulsatility more closely reflects intrinsic hypothalamic activation of GnRH neurons.”

Response: While in some proof-of-principle studies (see Suppl Fig. S1B and our response to point#2 of this referee) the use of GNX mice might be useful, we are afraid that for the sort of physiological analyses included in the body of our paper, comparison of GNX control and KiDKO mice might be misleading, as GNX in controls would evoke a massive activation of ARC Kiss1 neurons (but suppression of AVPV neurons), while in KiDKO mice, such activation is not likely taking place, likely due to the combination of primary defects in Kiss1 neurons and the suppressed levels of endogenous sex steroids (see also new Table 1 and our response to point-9 of referee#3). Hence, while we see the line of reasoning of the referee, we respectfully propose to keep the bulk of our analyses in gonadal-intact mice. We trust that incorporation of additional analytical data have helped to further characterize the phenotype of our model and satisfy the main demands of the referee. In any event, as we are sensitive to the issue raised by the referee, some comments on the use GNX animals have been included in the revised paper (see page 5, lines 125-126).

Minor comments

6. “The panels in Figure S4 could be re-ordered to align with the order in which they are discussed in the text (GnRH, GDX, KP, NMDA)”

Response: Done as suggested by the referee (see new Suppl. Fig. S4; and revised paper, at page 38, line 1174).

7. “There is no panel E in Figure S5 (it skips from D to F)”

Response: We regret the mistake; this has been corrected in the revised Suppl. Fig. S5 (and corresponding description of this figure).

Final comments

8. *“This is a very interesting and elegant manuscript in which targeted deletion of Dicer from Kiss1 neurons results in acquired loss of Kiss1 neurons and acquired hypogonadotropic hypogonadism, pointing to the importance of miRNAs in Kiss1 neurons and suggesting a role in acquired forms of central hypogonadism. The varying impacts on males and females, as well as on ARC vs AVPV Kiss1 neurons, further highlight sex differences in regulation of puberty and reproduction as well as distinctions between ARC and AVPV Kiss1 neuronal populations.*

This work is impactful. It is an editorial decision as to whether this is sufficient for publication in Nature Communications, or whether additional data (e.g., the mechanisms by which Dicer deletion results in hypogonadotropic hypogonadism, such as the miRNAs implicated) is warranted. Another question that is not fully answered is whether the acquired onset of hypogonadotropic hypogonadism is the result of a delay in the deletion of Dicer in this model, or whether Dicer is indeed deleted congenitally as proposed, but there is a delay in the impact on Kiss1 neuronal survival.”

Response: We truly appreciate the detailed evaluation and constructive comments of this referee. Based on our actions, described in detail above, we do believe that we have addressed the two main criticisms of this reviewer, related with the lack of mechanistic information and delineation of the time for Dicer ablation in our model. In parallel, we have also paid attention to all other comments of this evaluator. We trust that by these revisions, the referee and editor will find our work improved and publishable in the journal.

Reviewer #2

Comments

“The manuscript describes the effect of ablation of Dicer in Kiss1 neurons in the hypothalamus on the puberty and fertility in mice. The kisspeptin neuron number, the expression level of kiss1 gene, and the survival of kisspeptin neurons are all ruined by the conditional knockout of Dicer. The manuscript is interesting and well written, yet I only have some minor issues needed to be addressed detailed below”

Response: The positive comments of the reviewer on our work are greatly appreciated; our responses to all the specific comments of this referee are indicated below.

Specific comments

1. *“Line161-163, “adult KiDKO mice displayed a preserved number of GnRH-expressing neurons in POA (Fig. S3A, C)”, Why the inhibited kiss1 gene expression did not affect the expression of GnRH? Were there any other ways by which kisspeptin influenced puberty or fertility bypassing GnRH neurons? Could the author provide some other possible mechanisms mediating kisspeptin and puberty and fertility?”*

Response: The referee touches an interesting point that was possibly not sufficiently explained in our original submission. The available evidence from pharmacological, electrophysiological and functional genomic studies collectively points out that Kiss1 neurons in the hypothalamus primarily operate in GnRH neurons to centrally control reproductive function. While data supporting additional actions of kisspeptins in non-GnRH neurons have been presented, including findings from our group, these indirect central pathways are thought converge in physiological conditions, mostly, on GnRH neurons. In this scenario, I believe the most tenable explanation for the observed phenomenon, i.e., preserved GnRH mRNA expression despite markedly suppressed Kiss1 expression in ARC and AVPV, is that kisspeptin regulation of GnRH neuronal activity takes place at post-transcriptional levels, e.g., control of protein trafficking or secretion and/or control of electrical activity of GnRH neurons. In fact, our previous data conclusively demonstrated that central kisspeptin administration can evoke very potent LH (as surrogate marker of GnRH) responses without changes in GnRH mRNA expression (Navarro et al. Endocrinology 2005,146:156-63). This observation was in line with earlier reports suggesting that, in general terms, GnRH expression is scarcely controlled transcriptionally, but rather at the post-transcriptional level. Hence, while we cannot exclude the possibility of additional sites or pathways of action of kisspeptins, bypassing GnRH neurons, we think is reasonable to assume that such post-transcriptional regulation is key for these observations. Some comments on this issue have been included in the revised paper (see page 13, lines 389-390).

2. *“Fig S1A, what was the PCR template, what was the age of the DNA providers? I think it’s essential to clarify the time point of the Cre enzyme beginning to work, therefore, the age of mice when Dicer ablation in Kiss1 neurons came into being could be confirmed.”*

Response: This is a relevant point, which was also touched by referee#1. For specific details, see our response to point-2 of reviewer#1.

0. *“Line 210-214: The trends of increase of uterus and ovarian weights during 2w to 4w between the KiDKO and GoDKO female mice were different, what made the difference?”*

Response: In general terms, our findings are compatible with a more dramatic impact of Dicer ablation in GnRH neurons than in Kiss1 neurons during postnatal development, with suppressed ovarian and uterus weight gain during the pubertal transition in female GoDKO, but not KiDKO, mice. Our interpretation is that, while ablation of mature miRNA biogenesis in Kiss1 or GnRH neurons is deleterious for female fertility, KiDKO female are somewhat less sensitive to such adverse impact, at least during initial stages of pubertal maturation, due to the fact that they retain GnRH neurons, with conserved capacity to express GnRH and likely produce (at least to some extent) the neuropeptide. In contrast, the final output pathway is ablated in GoDKO mice, therefore resulting in a more dramatic phenotype. The above contention is briefly discussed in the revised version of the paper (see page 14, lines 398-401). Such a greater impact is also supported by our new dataset of sex steroid levels in KiDKO and GoDKO mice, included in response to point-9 of referee#3; for further details, see new Table 1 and page 6, lines 150-152, 161-162 and 166-167; and page 7, lines 198-200 and 203-206.

1. *“Why were the KiDKO female mice infertile though their ovulation seemed normal?”*

Response: This is a relevant issue that was only partially addressed in our initial submission. The fact that ovulation could be rescued by exogenous gonadotropin priming in KiDKO female mice was suggestive of a failure of the central mechanisms leading to the generation of the pre-ovulatory surge in KiDKO mice. This contention has been now explored specifically in our revised paper. Thus, while we could effectively induce a “pre-ovulatory” like LH surge in control mice properly primed with sex steroids (replicating the phenomenon of positive feedback driving the pre-ovulatory peak of gonadotropins), such induction was totally abrogated in adult KiDKO female mice (see new Fig. 2q), therefore supporting the idea that KiDKO mice have a primary defect at the central Kiss1-mediated pathways driving the ovulatory surge of GnRH/gonadotropins. This contention has been expanded and made clearer in the revised paper (see page 9, line 239-240; page 14, lines 406-407; and page 16, lines 481-482).

2. *“Line 241 “(Fig S6A,D, G)” could be “ (Fig S6A)”*.

Response: Revised and corrected, as suggested.

3. *“Before 4 weeks of age, the kisspeptin expression in the ARC was ruined by Dicer ablation more intensively than in the AVPV area, could you please explain such difference?”*

Response: This is a relevant point, which we also highlighted as potentially interesting based on our initial dataset. Our new mechanistic data have provided further insights into this phenomenon and actually can explain, at least partially, the basis for such differential impact. Thus, while our qPCR analyses in isolated Kiss1 neurons from the ARC of KiDKO mice evidenced an upregulation of three key transcriptional repressors (namely, Mkrn3, Cbx7 and Eap1) of Kiss1 expression in the arcuate, this phenomenon was not detected in Kiss1 neurons from the AVPV. This finding strongly suggests that the impact of disrupted mature miRNA biogenesis is different between ARC and AVPV Kiss1 neurons, therefore surfacing an interesting divergence in the physiological relevance of miRNA-dependent regulation of Kiss1 in these distinct neuronal populations. This contention, which we believe is of considerable relevance, has been discussed in detail in our revised paper (see page 2, lines 41-42 and 45-46; page 11, lines 314-331; and page 12, lines 353-365, cont. page 13, lines 366-383).

4. *“Dicer ablation in kisspeptin neurons suppressed the number of kisspeptin neurons and decreased the survival of these neurons after 4 weeks according to Fig3 and Fig 6. Could it be concluded that the survival of kisspeptin neurons depended on or independent of the kisspeptin expression?”*

Response: While this possibility cannot be totally excluded on the basis of our current dataset, we do believe that the compromise in survival of Kiss1 neurons occurs via mechanisms that are, to a large extent, kisspeptin-independent. This is supported, among other findings, by the time-lag between the observed drop in kisspeptin immunoreactivity and Kiss1 neuronal survival (assessed by YFP reporter-tagging of Kiss1 neurons; see Figure 6). In the same vein, our independent study, coming from a model of congenital ablation of Kiss1 from KNDy (Tac2-positive) cells, documents that NKB-expressing cells are fully preserved despite complete ablation of kisspeptin production. In any event, we would like to note that this is a totally independent study, which aims to address completely independent goals (i.e., to dissect out the physiological roles of kisspeptins from KNDy vs. non-KNDy cells in the physiological control of reproduction). For this reason, we

have refrained to discuss on this dataset in the present paper. Anyhow, we have succinctly addressed this issue in the revised paper (see page 10, lines 301-302).

8. “*Lin2 532-536: In “material and methods” section, pubertal maturation were evaluated by body weight monitoring, assessment of BPS (in male) and VO (in female). However, the results of BPS and VO of KiDKO mice were not provided. Were there any differences in these pubertal phenotypes between KiDKO and control mice?”*”

Response: This piece of information was provided in original figures, see Suppl. Fig 5S, and was briefly mentioned in the text of the result section. Anyhow, as this piece of info might have not been presented in a sufficiently clear manner in the original submission (e.g., we did not refer to the abbreviation of BPS and VO), description of this dataset has been minimally amended and made clearer in the revised version of our paper (see page 7, lines 195-196 and page 8, lines 207-208).

Reviewer #3

Comments

“In this MS the Authors have generated mice with a deletion of Dicer in Kiss1 expressing cells and analyzed the neuroendocrine and reproductive phenotypes during different stages of HPG axis development and maturation. For some of the presented data, they compared Dicer cKO in Kiss1 neurons with Dicer cKO in GnRH neurons. While the phenotypic analysis is extensive and well done, the MS suffers from lack of any mechanistic data that support their conclusions. Surprisingly, the Authors have not included the extent to which Dicer is deleted in the specific cell populations.”

Response: We are sensitive to the main points of this referee, which have been also touched by the editor and the other evaluators. As mentioned in our response to the editor, our review efforts have focused mainly in addressing in detail the mechanistic basis of the phenotype and in the validation of Dicer ablation in Kiss1 neurons in our KiDKO model. These analyses have allowed us to significantly enlarge the experimental data included in our study, providing a mechanistic insight of the observed phenotype and reinforcing the validity of our model to reach solid conclusions. For further details, see our responses to the specific comments of this referee (below).

Specific additional comments

1. *“The title is confusing. Could be reworded and also the Authors may want to consider to include Dicer deletion in GnRH cells also”*

Response: In the title, we aimed to provide a direct message of the major output of the study, namely, the importance of miRNA-biogenesis in Kiss1 neurons for puberty and fertility, which is greater in females. While in our study we included, for comparative purposes, some data from GoDKO mice that are interesting and informative, we do believe that highlighting this piece of information in the title might be misleading, as our intention for showing GoDKO data was just as a means to illustrate differences between Kiss1 and GnRH neurons in their dependence of miRNA-biogenesis and functional roles in the control of the reproductive axis. For this reason, we respectfully propose not to include any mention of GnRH neurons in our title. In the same vein, after some attempts, we have not found an improved version of our title, which we propose to keep in its original version. But we would be open to reconsider specific suggestions of the referee or editor to improve the title, if they consider this mandatory.

2. *“It would be nice to show how much Dicer is deleted in the Cre+ cells by qPCR, similarly in FACS sorted cell populations, it would be important to show the levels of Kiss1, instead of qualitatively showing in situ expression data”*

Response: This is a relevant point, which was also touched by referee#1. Based on these suggestions, we have conducted qPCR analyses in isolated Kiss1 neurons from our KiDKO mice, of both sexes, and have fully confirmed, in an unambiguous manner, that Dicer expression is markedly suppressed in our conditional model. In addition, our qPCR analyses in FACS Kiss1 cells confirmed detectable, albeit partially suppressed expression of Kiss1 in peripubertal KiDKO mice. Anyhow, we would like to stress that, apart from the BaseScope data, all in situ hybridization analyses of Kiss1 expression, conducted in KiDKO mice at different age-points, provided quantitative data. For further details, see also new Suppl. Fig. S1 and page 5, lines 131-135; and page 37, lines 1121-1124, of the revised paper. See also our response to point-2, reviewer#1.

3. *“The Authors have not mentioned anything about which miRNAs target Kiss1 mRNA and if these are suppressed or increased or unchanged in Dicer cKO mice in Kiss+ cells in both ARC and AVPV populations”*

Response: The referee touches on an important issue that has been also addressed in point-4 of referee#1. Our line of reasoning for these mechanistic analyses was that, since Dicer ablation should remove all mature miRNAs in a cell-specific manner, high-throughput detection of the miRNAs present, differentially-expressed or absent in Kiss1 neurons from KiDKO mice might be less informative. Similarly, since miRNAs are predominantly repressors of mRNA stability and/or translation, and Kiss1 mRNA and protein were actually suppressed in KiDKO mice, we assumed that ablation of direct miRNA regulators of Kiss1 gene are less likely to contribute to this molecular phenotype, but rather elimination of miRNA repressors of transcriptional repressors of Kiss1 would be likely to play a major role.

On this basis, we screened the literature and selected a panel of reported transcriptional regulators of Kiss1, including putative repressors or activators. For details of the major transcriptional repressors and activators of Kiss1 selected for our analyses, see our revised paper (page 11, lines 318-322). Of note, these qPCR analyses were conducted in isolated Kiss1 neurons obtained by FACS from our KiDKO mouse line, thus providing a notable level of precision to our studies.

Remarkably, these expression analyses have allowed us to delineate a tenable mechanism for the observed suppression of Kiss1 in our KiDKO mouse, as major determinant for its phenotype, since three important transcriptional repressors of Kiss1, namely Mkrn3, Cbx7 and Eap1, were found to be upregulated in male and female KiDKO mice. Of note, in a recent unrelated study, we have documented that Mkrn3 is negatively regulated by miR-30b (Heras et al., PLoS Biology 2019;17(11):e3000532), which also has a putative seed region in the 3'-UTR of Cbx7 (new bioinformatic analyses included in the revised paper; see page 29, lines 845-849). Therefore, our present study does not only disclose the need of proper miRNA biosynthesis for adequate functioning of Kiss1 neurons, but demonstrate also the physiological relevance of novel miRNA-regulatory pathways of Mkrn3 and Cbx7, as well as possibly Eap1, in the control of Kiss1 expression, as principal component for acquisition of reproductive capacity.

We do believe that addition of this relevant piece of data endows our paper with a substantial mechanistic dimension, in line with the request of the referees and editor. Importantly, analyses have differentially considered ARC vs. AVPV populations, as mentioned also by this referee. Overall, these issues have been extensively discussed in the revised paper; see page 2, lines 41-42 and 45-46; page 11, lines 314-331; and page 12, lines 353-365, cont. page 13, lines 366-383.

4. *“Line 112: What do the Authors mean “fine control of the reproductive brain”?”*

Response: We intended to refer to brain mechanisms responsible for the precise regulation of reproductive function. We admit this might be misleading; we have rephased, following the indications of the referee (see page 4, lines 109-110).

5. *“Line 117: it should be Mendelian ratio; not Mendelian rate”*

Response: Thank you for picking this typo; this has been corrected accordingly (see page 5, line 115).

6. *“Line 136 & Line 138: shouldn't it be conditional null animals? And conditional null females? Not null animals and null females!”*

Response: The referee is totally right; this has been amended accordingly (see page 6, lines 143 and 145; and also in page 9, line 246; page 10, line 288; page 15, lines 456 and 459; and page 22, lines 627 and 643).

7. *“At several places in the Results and Discussion, the Authors have made qualitative statements without actually mentioning how much suppression or increase was noted in expression of genes and whether these changes were statistically significant. These need to be fixed throughout the text”*

Response: We see the point of the referee and have carefully re-checked the Results and Discussion sections to avoid such statements. Indeed, we do believe that some of the revisions inserted in the new version of the paper have helped very much us in this effort. For instance, in our original submission, we alluded to suppressed expression of let-7b as proxy marker of effective ablation of Dicer in Kiss1 neurons, based on a semi-quantitative analysis of FAC sorted Kiss1 cells from one representative KiDKO mouse. In the new version, this former dataset has been replaced by real-time quantitative analyses using qPCR in isolated Kiss1 cells to document in a very precise manner not only the suppression of let-7b levels but also the actual marked reduction of Dicer mRNA expression in Kiss1 neurons from KiDKO mice. Similarly, we have provided additional direct experimental support to other statements, such as the lack of positive feedback and ovulatory failure or the actual hypogonadal state of our model, now documented by direct measurements of testosterone, progesterone or estradiol in male and female KiDKO mice. Finally, our educated speculation about the potential involvement of transcriptional repressors of Kiss1 in the molecular phenotype of KiDKO mice has been now substantiated by direct qPCR measurements in isolated Kiss1 neurons. We trust that these additional datasets, and the statistical analyses associated to them, have

strengthen our conclusions and satisfied the point made by this referee. For specific actions on the issues mentioned above, see our responses to the editor and referees.

5. *“Line 179: could this be a direct pituitary effect?”*

Response: We assume the reviewer refers here to the responses to NMDA, Kp-10, GnRH and even GNX in KiDKO and GoDKO mice. While we cannot completely rule out some marginal compromise at the pituitary level, analysis of the responses to GnRH, which actually operates at the pituitary, points to a major hypothalamic compromise, related with dysfunction of either Kiss1 or GnRH neurons in our KiDKO and GoDKO models, respectively. Thus, both models equally responded to a single bolus of GnRH, with near-to-control responses. However, GoDKO mice were totally irresponsive to Kp-10, which acts at GnRH neurons, while KiDKO mice retained responsiveness to Kp-10, as in the latter model, GnRH neurons seemingly maintain functionality. Admittedly, the magnitude of responses to GnRH (in both models) or Kp-10 (in KiDKO mice) was not maximal, but our interpretation is that the expected decrease in GnRH output in both KiDKO (due to the lack of endogenous kisspeptin drive) and GoDKO (due to dysfunction of GnRH neurosecretion) mice should have resulted in insufficient pituitary self-priming, which may partially reduce the magnitude of LH responses to a single bolus of the various stimuli. But this decrease would be secondary to hypothalamic dysfunction and not caused by alterations primarily at the pituitary level. Some brief comments about this phenomenon can be found in the revised manuscript (page 7, lines 181-183).

6. *“Are serum levels of steroids measured (estradiol, progesterone and testosterone) in cKO mice at different ages?”*

Response: This piece of information was not included in our original submission. Following the indications of the referee, we have conducted measurements, using up-to-date mass spectrometry methods, of relevant sex steroids in male and female KiDKO mice and their respective controls, at two relevant age points (peripuberty and adulthood). The results are included in new Table 1, and have been described in detail in the revised paper (see page 6, lines 150-152, 161-162 and 166-167; and page 7, lines 198-200 and 203-206). This new dataset fully confirms the neuroendocrine phenotype of our model, and we believe is a valuable addition to our paper, made in response to referee’s suggestion.

0. *“The Authors argue repressors are activated in deleted cells and therefore Kiss1 protein was suppressed. But, there is no data provided to support this. It is essential for the Authors to check RNA/protein expression of repressors or activators and miRNAs that target these in Dicer deleted cells to come up with any mechanistic explanation for the phenotypes they observed.”*

Response: While our assumption was made on the basis of educated hypotheses, we totally agree this was an speculation. As described also in our responses to the editor and the other referees (see our response to point-4 of referee#1), we have undertaken expression analyses in isolated Kiss1 neurons from KiDKO mice to specifically address this issue. Our new dataset, presented in the Figure 7 of the revised manuscript, clearly documents that global ablation of mature miRNA biogenesis in Kiss1 neurons causes an upregulation of the expression of three reported Kiss1 repressors, namely, Mkrn3, Cbx7 and Eap1, which are likely to contribute to the changes in Kiss1/kisspeptin levels seen in our conditional null model.

As mentioned in our response to point-4 of referee#1, we believe that addition of this relevant piece of data endows our paper with a substantial mechanistic dimension, in line with the request of the referees and editor. Accordingly, these issues have been extensively discussed in the revised paper: see page 2, lines 41-42 and 45-46; page 11, lines 314-331; and page 12, lines 353-365, cont. page 13, lines 366-383.

Reviewers' Comments:

Reviewer #1:

Remarks to the Author:

The authors have addressed the questions and concerns raised in the initial review to my satisfaction. In particular, they have now documented early, congenital ablation of Dicer in Kiss1 neurons, evident by PND1, and confirm ablation of Dicer from Kiss1 neurons by PND14 and PND28 by FACS sorting. In addition, they now demonstrate that several putative repressors of Kiss1 were upregulated in FACS-sorted Kiss1 neurons of KiDKO mice at PND28, suggesting a potential mechanism of the effects of Dicer deletion in Kiss1 neurons.

Reviewer #3:

None

Reviewer #4:

Remarks to the Author:

In general, Roa et al revised the original manuscript well based on the comments raised by the reviewers. However, there is one issue that is not clear yet even after revision for the reviewer 2 (comment 4).

As commented by the other reviewers (comment 4 of reviewer 1 and comment 3 of reviewer 3), the original manuscript did not provide evidence for molecular mechanism(s) by which Dicer ablation affects Kisspeptin expression, leading to the following phenotypes shown in the manuscript. It is very important to demonstrate which microRNAs are affected by the ablation of Dicer in Kiss1 neuron for understanding the molecular mechanisms for the phenotypes.

To respond these comments from all 3 reviewers, the authors repeatedly insisted that Dicer ablation leads to complete removal of mature microRNAs in a cell-type specific manner so that high-throughput screening of microRNAs in Kiss1 neurons without Dicer is not informative. However, there are many references that deletion of Dicer or other key factors for microRNA biogenesis did not lead to complete removal of microRNAs. In fact, these references showed that only a few dozens of microRNAs are reduced in cells where Dicer or other key factors are deleted. I assume that it is critical to provide a list of microRNAs whose expression is significantly altered by the ablation of Dicer in Kiss1 neuron. The author responses to the comments raised by all three reviewers simply provided descriptive data that may not be directly associated with reduced expression of Kiss1 in Kiss1 neuron. Although the authors mentioned miR-30b that negatively regulates the expression level of transcriptional repressor Mkrn3, it is a speculation from the literature and there is no direct evidence that this event actually occurs in Kiss1 neuron deficient of Dicer. To provide direct evidence for the molecular mechanism for the phenotypes shown in the submitted manuscript, Figure 7 should have been started from the identification of microRNAs whose expression is significantly affected by the deletion of Dicer in this neuron. Molecular association between altered microRNAs and transcription regulators of Kiss1 gene should have been elucidated instead of selecting putative microRNA targets and their following target mRNAs that may affect Kiss1 gene expression from the literature.

Editor Comments

1. "Your manuscript entitled "Female-biased dependence of puberty and fertility on microRNA biogenesis in Kiss1 neurons" has now been seen again by referees 1 and 3. Reviewer 2 was not available, so we recruited a new referee (#4) to comment on your revision. You will see from their comments below that ... suggest further studies to address which microRNA are specifically affected by Dicer ablation. I discussed these suggestions with the editorial team, and also further consulted with reviewer 1. While we feel that the proposed experiments would enrich the work, **they will not be necessary for further consideration of the manuscript**, also in consideration of the work already presented and of the time that might be required for the additional experiments.

Therefore, **I am delighted to say that we are happy, in principle, to publish a suitably revised version** in Nature Communications under the open access CC BY license (Creative Commons Attribution 4.0 International License).

We invite you to revise your paper one last time **to address the concerns of reviewer 4 with a caveat/discussion** in the text, and our **editorial requests in the attached document(s)**.

Response: We have followed your indications as to "revise your paper one last time to address the concerns reviewer 4 with a caveat/discussion in the text". We have done so, as you can see in the revised MS file, page 12, lines 348-352.

While we see the point of the referee, we do believe that this approach, which is technically very demanding, would not enable us to identify specific miRNAs responsible for the phenotype of our KiDKO mouse. In contrast, our targeted approach permitted us to define a tenable pathway to explain a global mechanisms for Kiss1 repression by canonical miRNA biogenesis in Kiss1 neurons. This aspect has been now discussed in the revised paper (page 12, lines 348-352). We do appreciate your support on this issue.

In addition, we have carefully considered all the editorial requests, and edits/amendments have been inserted in the revised version of the MS, accordingly. These have been indicated in the Author Checklist (provide) and highlighted in the red-marked version of the revised MS, included in the new submission.